# Towards Robustness of Person Search Against Corruptions

## Abstract

Person search aims to simultaneously detect and re-identify a query person within an entire scene, involving detection and re-identification as a multi-task problem. While existing studies have made significant progress in achieving superior performance on clean datasets, the challenge of robustness under various corruptions remains largely unexplored. To address this gap, we propose two benchmarks, CUHK-SYSU-C and PRW-C, designed to assess the robustness of person search models across diverse corruption scenarios. Previous studies on corruption have been conducted independently for single tasks such as re-identification and detection. However, recent advancements in person search adopt an end-to-end multi-task learning framework that processes the entire scene as input, unlike the combination of single tasks. This raises the question of whether independent achievements can ensure corruption robustness for person search. Our findings reveal that merely combining independent, robust detection and re-identification models is not sufficient for achieving robust person search. We further investigate the vulnerability of the detection and representation stages to corruption and explore its impact on both foreground and background areas. Based on these insights, we propose a foreground-aware augmentation and regularization method to enhance the robustness of person search models. Supported by our comprehensive robustness analysis and evaluation framework our benchmarks provide, our proposed technique substantially improves the robustness of existing person search models. Code will be made publicly available.

## 1 Introduction

Person search is a task that involves detecting individuals in complex scenes and subsequently re-identifying the same individuals from a gallery of scene images. Significant advancements in this task have been made through the discriminative capabilities of deep neural networks (DNNs). Since person search requires both pedestrian detection and person re-identification simultaneously, earlier studies have adopted a two-step framework (Lan et al., 2018; Chen et al., 2018; Han et al., 2019; Wang et al., 2020), where independent detection and re-identification models are applied sequentially. More recent models have adopted a one-step approach (Li & Miao, 2021; Lee et al., 2022; Cao et al., 2022; Yu et al., 2022), which processes the entire scene in a single pass and leverages joint end-to-end multi-task learning, leading to significant improvements.

Despite these advancements, DNNs have shown vulnerability to common corruptions such as noise, blur, and compression artifacts, often resulting in significant performance degradation (Liu et al., 2024a; Chen et al., 2024; Kong et al., 2024; He et al., 2023; Schiappa et al., 2022; Yi et al., 2021). This vulnerability leads to the necessity for models to maintain robustness under such challenging conditions, driving numerous studies focused on enhancing corruption robustness (Mintun et al., 2021; Kar et al., 2022; Dooley et al., 2022; Dong et al., 2023). This line of research has been explored in the fields of detection (Michaelis et al., 2019; Mao et al., 2023; Lee et al., 2024) and re-identification (Chen et al., 2021). However, to our knowledge, the impact of data corruption on person search models remains largely unexplored, highlighting the need for further investigation in this area.

To tackle this issue, we introduce two new benchmarks – CUHK-SYSU-C and PRW-C – that extend existing popular person search datasets (Xiao et al., 2017; Zheng et al., 2017) to address their lack

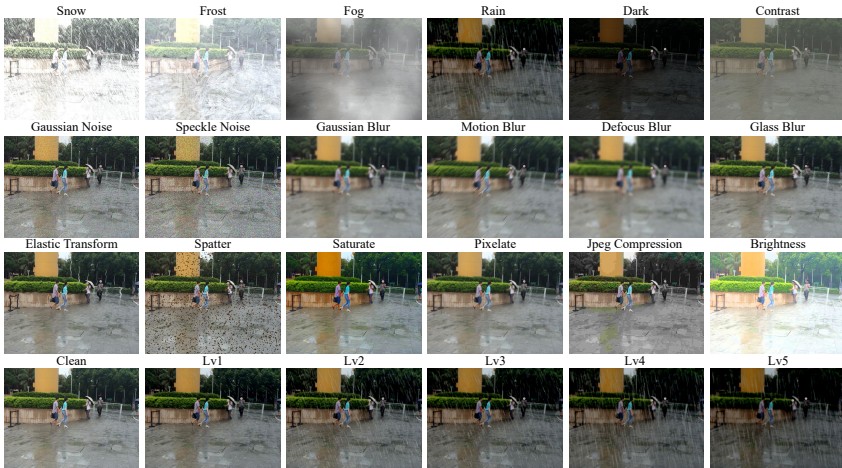

Figure 1: **Examples of corruption types with varying severity** in our proposed benchmark.

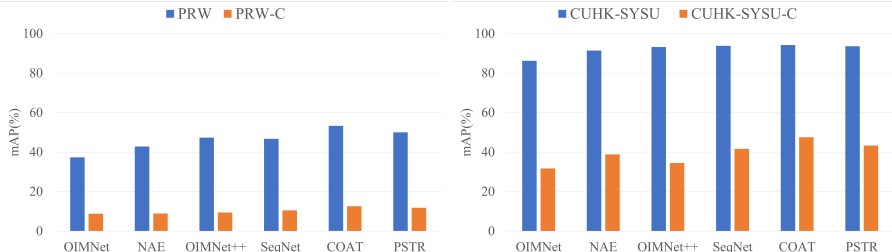

Figure 2: **Performance evaluation of SoTA person search models on the proposed corruption benchmarks: CUHK-SYSU-C and PRW-C.** We employ six state-of-the-art person search models: OIMNet (Xiao et al., 2017), NAE (Chen et al., 2020), OIMNet++ (Lee et al., 2022), SeqNet (Li & Miao, 2021), COAT (Yu et al., 2022), PSTR (Cao et al., 2022). Models are trained on CUHK-SYSU and PRW, then evaluated on the proposed benchmarks CUHK-SYSU-C and PRW-C, respectively.

of robustness evaluation under corrupted conditions. These benchmarks incorporate 18 types of corruption with five severity levels, enabling a detailed evaluation under diverse corruption scenarios, as shown in Figure 1. Given these proposed benchmarks, we extensively evaluate the robustness of corruption in seminal state-of-the-art person search models. Moreover, we explore a straightforward solution that naturally arises: integrating a robust detection model (Lee et al., 2024) with a re-identification model (Chen et al., 2021), both of which are designed for corruption robustness. However, our experimental results reveal that existing person search models remain highly vulnerable to corruption, and a simple integration approach is insufficient to address this issue (Section 3.1). For instance, as shown in Figure 2, we observe a notable performance drop in existing person search models, with relative mAP declines of up to 80%.

To investigate the underlying reasons for this phenomenon, we conduct further analysis of corruption vulnerability, considering the unique aspects of this task. Since person search models typically localize pedestrian candidates through a detection head and then extract person representations from the detected regions, we explore the sensitivity of the detection and re-identification stages to corruption (Section 3.2). Additionally, given the complex scene inputs and large receptive fields in person search models, we assess how corrupted regions in the input images, including the foreground (containing the person) and the background, affect performance (Section 3.3). Our experimental findings indicate that the re-identification stage and the foreground regions are particularly vulnerable to corruption, leading to significant performance drops.

To this end, we propose a simple yet effective method to enhance corruption robustness—foreground-aware augmentation and regularization for robust person representation, which can be easily applied to various existing person search models. Specifically, we apply selective augmentations to the foreground in the input scene images and compute a regularization between the person representations of the original and foreground-augmented images to improve robustness against corruption. Thanks to its simple and easy-to-implement design, extensive experimental results on

CUHK-SYSU-C and PRW-C demonstrate that our method consistently improves the corruption robustness of five state-of-the-art person search models.

In summary, the contributions of this paper are as follows:

- We propose new benchmarks, CUHK-SYSU-C and PRW-C, to evaluate corruption robustness and reveal the significant vulnerability of state-of-the-art person search models to corruption.
- We analyze the sensitivity of both the detection and re-identification stages to corruption and explore the impact of corrupted foreground and background regions.
- Based on this analysis, we propose a foreground-aware augmentation method and a regularization specifically designed to enhance robust person representation.
- Extensive experiments on CUHK-SYSU-C and PRW-C demonstrate that our method significantly improves the corruption robustness of five state-of-the-art person search models.

## 2 RELATED WORKS

**Person Search.** Existing person search works are typically categorized into two approaches: the two-step and the one-step methods. Two-step approaches involve two independent detection and reID models, where a detection model first detects people appearing in the scene, and then a reID model recognizes the detected individuals. The primary goal of this two-step approach (Han et al., 2019; Dong et al., 2020; Wang et al., 2020) is to specialize these models, which were initially designed for independent tasks, for the person search task by employing techniques like adapting the prediction by a detector to facilitate the reID process. On the other hand, the one-step approach (Xiao et al., 2017; Munjal et al., 2019; Kim et al., 2021; Zhang et al., 2021a; Han et al., 2021; Li & Miao, 2021; Yu et al., 2022; Lee et al., 2022; Cao et al., 2022; Li et al., 2022; Han et al., 2022; Yan et al., 2023), which performs both sub-tasks with a single model, has recently gained popularity due to its simplicity and efficiency, reducing the number of parameters by nearly half.

One-step approaches take the entire scene as input and jointly train the detection and reID heads in an end-to-end manner. This is commonly achieved by employing a Faster R-CNN (Ren et al., 2016) detection head and reID head with the OIM (Xiao et al., 2017) loss. The one-step approach focuses on addressing challenges that arise from the joint learning of the two tasks. Chen et al. (2020) address conflicting learning objectives, and SeqNet (Li & Miao, 2021) and OIMNet++ (Lee et al., 2022) incorporate the detection quality on reID head training and search performance. PSTR (Cao et al., 2022) and COAT (Yu et al., 2022) enable recent one-step person search frameworks to leverage the advantages of transformers. Another challenge unique to person search, unlike conventional reID, is the influence of background information on extracting person representations from larger receptive fields. GLCNet (Qin et al., 2023) mitigates this by extracting features from the entire global scene, including both the background and foreground, to improve the final person embedding. While these studies have made significant advancements in the person search field, they have not been sufficiently investigated for corruption robustness.

**Benchmarking Robustness to Corruption.** Deep neural networks are often considered generalizable, but they are not as robust to corruption as humans are (Leveque et al., 2022). The pioneering study by Hendrycks & Dietterich (2019) introduced a benchmarking paradigm, where synthetic corruptions are used to evaluate model robustness. This benchmark, ImageNet-C, employs algorithmically generated corruptions, revealing the unexpected vulnerability of networks to even simple perturbations. Following this paradigm, robustness research has expanded into broader areas of vision tasks, including object detection (Michaelis et al., 2019), pose estimation (Wang et al., 2021), semantic segmentation (Kamann & Rother, 2020), person re-identification (Chen et al., 2021), and depth estimation (Kong et al., 2024).

While significant progress has been made in improving the robustness of specific vision tasks, existing research primarily focuses on individual tasks (Wang et al., 2021; Kamann & Rother, 2020; Kong et al., 2024). Person search, however, is a multi-task problem involving both detection and re-identification, each requiring different input types and resulting in conflicting optimization (Lin et al., 2021). Although previous studies (Michaelis et al., 2019; Chen et al., 2021) have contributed substantially to our understanding of how models for individual tasks perform under various corruption scenarios, the robustness of the multi-task framework in person search against corrupted environments has yet to be sufficiently explored.

Table 1: **Performance of state-of-the-art person search models under corruption.** The column 'CUHK-SYSU' ('PRW') denotes the performance measured on the clean images, and 'CUHK-SYSU-C' ('PRW-C') indicates the performance measured on the corrupted images.

| Method | CUHK-SYSU | | | | CUHK-SYSU-C | | | | | | | |
| --- | --- | --- | --- | --- | --- | --- | --- | --- | --- | --- | --- | --- |
| | Search | | Detection | | Search | | | | Detection | | | |
| | R@1 | mAP | Recall | AP | rR@1 | rmAP | R@1 | mAP | rRecall | rAP | Recall | AP |
| OIMNet | 87.7 | 86.2 | 87.5 | 81.1 | 37.7 | 36.7 | 33.0 | 31.7 | 78.8 | 77.4 | 68.9 | 62.7 |
| NAE | 92.3 | 91.4 | 92.3 | 86.9 | 44.3 | 42.4 | 40.9 | 38.8 | 72.7 | 71.8 | 67.1 | 62.3 |
| OIMNet++ | 94.0 | 93.2 | 92.4 | 88.9 | 38.6 | 37.0 | 36.3 | 34.5 | 77.3 | 76.3 | 71.4 | 67.8 |
| SeqNet | 94.5 | 93.8 | 92.0 | 89.2 | 46.2 | 44.4 | 43.6 | 41.6 | 74.5 | 74.1 | 68.6 | 66.1 |
| COAT | 94.7 | **94.2** | 91.3 | 88.1 | **52.6** | **50.4** | **49.8** | **47.5** | 76.4 | 75.9 | 69.8 | 66.9 |
| PSTR | **94.9** | 93.6 | 89.5 | 66.9 | 49.5 | 46.3 | 47.0 | 43.3 | 71.6 | 74.1 | 64.0 | 49.6 |
| OADG+CIL | 94.0 | 92.9 | **94.7** | **90.9** | 48.5 | 46.0 | 45.6 | 42.8 | **89.8** | **88.4** | **85.0** | **80.4** |

| Method | PRW | | | | PRW-C | | | | | | | |
| --- | --- | --- | --- | --- | --- | --- | --- | --- | --- | --- | --- | --- |
| | Search | | Detection | | Search | | | | Detection | | | |
| | R@1 | mAP | Recall | AP | rR@1 | rmAP | R@1 | mAP | rRecall | rAP | Recall | AP |
| OIMNet | 76.7 | 37.3 | 93.7 | 85.0 | 45.8 | 23.6 | 35.2 | 8.8 | 79.4 | 78.8 | 74.4 | 66.9 |
| NAE | 80.6 | 42.8 | 93.3 | 88.7 | 42.6 | 20.9 | 34.3 | 8.9 | 67.3 | 64.9 | 62.8 | 57.6 |
| OIMNet++ | 83.2 | 47.3 | **96.3** | 93.2 | 43.9 | 19.9 | 36.5 | 9.4 | 75.2 | 74.2 | 72.4 | 69.2 |
| SeqNet | 83.4 | 46.7 | **96.3** | **93.9** | 46.7 | 22.6 | 38.9 | 10.5 | 74.3 | 73.6 | 71.5 | 69.1 |
| COAT | 87.4 | **53.3** | 94.9 | 92.6 | 49.9 | 23.7 | 43.6 | **12.6** | 67.1 | 66.6 | 63.6 | 61.7 |
| PSTR | **88.1** | 50.0 | 90.4 | 77.7 | 51.0 | 23.7 | **44.9** | 11.8 | 69.9 | 61.7 | 63.2 | 47.9 |
| OADG+CIL | 85.6 | 41.6 | 91.7 | 89.4 | **52.0** | **27.5** | 44.5 | 11.5 | **94.6** | **93.8** | **86.7** | **83.9** |

## 3 ROBUSTNESS ANALYSIS ON PERSON SEARCH

### 3.1 INVESTIGATION OF CORRUPTION ROBUSTNESS ON PERSON SEARCH

**Benchmark Dataset Design.**    To evaluate the robustness of person search models, we propose two benchmarks: CUHK-SYSU-C and PRW-C. These benchmarks are built upon the widely adopted CUHK-SYSU (Xiao et al., 2017) and PRW (Zheng et al., 2017) datasets, which have been extensively used in numerous person search studies (Han et al., 2019; Chen et al., 2018; Li & Miao, 2021; Kim et al., 2021). CUHK-SYSU dataset offers a diverse range of backgrounds from various urban scenarios, CUHK-SYSU-C retains this diversity while introducing corrupted scenes captured in those environments. The PRW dataset includes images captured from six distinct viewpoints at each location, and PRW-C preserves this multi-view feature, adding corruption across various angles. For evaluation using our benchmarks, we utilize the same test splits from these datasets.

We extend the corruptions used in the pioneering study by Hendrycks & Dietterich (2019) to the person search task. Considering the characteristics of urban outdoor environments in person search, we incorporate rain (Chen et al., 2021) and dark (Kong et al., 2024) as corruption types of our benchmarks. The corruption types of our benchmark include: 'gaussian noise', 'speckle noise', 'defocus blur', 'glass blur', 'motion blur', 'gaussian blur', 'snow', 'frost', 'fog', 'brightness', 'spatter', 'rain', 'dark', 'contrast', 'elastic', 'pixelate', 'jpeg compression', and 'saturate'. Each corruption is applied at five distinct severity levels, with higher severity indicating greater image degradation (*i.e.*, more severe corruption). The severity levels are established based on the traits of corruption scenarios observed in real-world. As an example, the rain corruption depicted in Figure 1 has various attributes including slope of rain droplets, color of rain droplets, drop length & width, overall blurriness and brightness to simulate different severity. Details for all corruptions are available in Appendix.

In person search, the goal is to match a given query image to its corresponding gallery image in a database; corruption can be applied to either the query or the gallery image. Query and gallery images may not always exhibit the same type or severity level of corruption. We construct the benchmarks by randomly applying different corruption types and severity levels to query and gallery images, similar to prior work in instance retrieval (Chen et al., 2021). We repeat the evaluation process five times and report the mean performance for experimental results. For benchmark statistics and further discussion of this design choice, see Appendix C and B, respectively.

**Evaluation Models.**    To explore the corruption robustness of person search models, we employ six seminal state-of-the-art models including both CNN and transformer architectures: OIMNet (Xiao et al., 2017), NAE (Chen et al., 2020), OIMNet++ (Lee et al., 2022), SeqNet (Li & Miao, 2021), COAT (Yu et al., 2022), and PSTR (Cao et al., 2022). More details of these models are provided

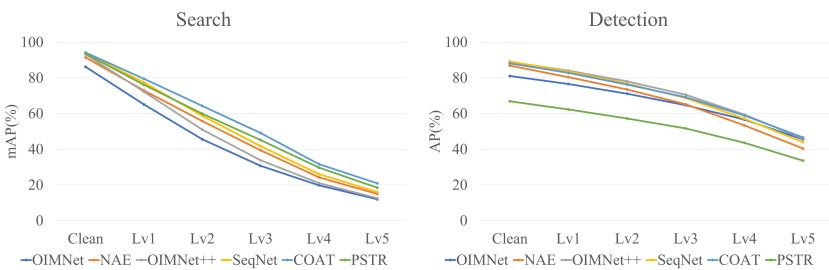

Figure 3: **Person search model performance across various corruption severities**, evaluated on the CUHK-SYSU-C benchmark.

in the appendix. For NAE, OIMNet++, SeqNet, PSTR, and COAT, we utilize the official checkpoints and configurations provided by the authors. Considering the evolution of training techniques since OIMNet was published, we re-implement and report the results. We first train the model on the training splits of CUHK-SYSU and PRW, then evaluate it on the CUHK-SYSU-C and PRW-C benchmarks. Unless otherwise specified, SeqNet is used for most experiments, as it serves as a baseline in several preceding works (Li et al., 2022; Jaffe & Zakhor, 2023; Li et al., 2023), thanks to its simple yet effective architecture.

**Evaluation Metrics.** Given that person search inherently involves both detection and reID, we report the performance metrics in two categories: detection and search performance. For search performance, we adopt the widely used metrics of mean Average Precision (mAP) and Cumulative Matching Characteristic at Rank-1 (R@1). For detection performance, we utilize Recall and Average Precision (AP). Following existing corruption studies (Michaelis et al., 2019; Wang et al., 2021; Schiappa et al., 2022), we also evaluate the relative performance drop caused by corruption, comparing the performance on the corrupted set to the clean set. This includes *relative mAP* (rmAP), *relative Rank-1* (rR@1), *relative Recall* (rRecall), and *relative AP* (rAP). For instance, relative mAP (rmAP) is calculated by dividing the mAP from the corrupted set by the mAP from the clean set (*i.e.*, rmAP = 'mAP from corrupted set'/'mAP from clean set').

**Evaluation Results.** We evaluate the performance of state-of-the-art person search models under corruption. Table 1 presents the performance of each model on both CUHK-SYSU (PRW) and CUHK-SYSU-C (PRW-C). The results show a significant drop in performance under corruption, with up to an 80% decline in mAP on PRW-C compared to the clean set. This reveals the vulnerability of current person search models to corruption and highlights the necessity of developing corruption-robust person search models. We also evaluate these models across five different severity levels of corruption. As shown in Figure 3, both the search and detection capabilities of person search models decrease as the severity level increases. Specifically, we observe a 20% to 30% decline in search performance with each increase in severity, while detection performance degrades at a more gradual pace. Performance begins to degrade from the first severity level, and when severity reaches level 5, overall search performances drop to approximately 10% of mAP.

**Evaluation of Combining Existing Robustness Methods.** In the last row of the Table 1, we further investigate whether combining existing corruption-robust detection and reID models can achieve robust person search. Specifically, we combine two independent models: OADG (Lee et al., 2024), a robust detection model designed for corrupted environments, and CIL (Chen et al., 2021), a reID model built for re-identification under corruption scenarios. For a fair comparison, we train both models on the person search dataset. Specifically, OADG and CIL are initialized with ImageNet (Deng et al., 2009) pre-trained ResNet-50 (He et al., 2016) and trained on the clean CUHK-SYSU (PRW) dataset. For evaluation, OADG is first used to detect pedestrians on CUHK-SYSU-C (PRW-C), and the detected individuals are then input into CIL to extract person representations, following the standard protocol of the two-step approach. The results in Table 1 demonstrate that this simple integrating approach is not sufficient to achieve robustness to corruption. Although it shows the best rR@1 and rmAP on PRW-C, its overall performance on the corrupted benchmarks, especially in R@1 and mAP, remains less effective compared to other person search models.

In summary, these results indicate that, despite the corruption-robust design of these methods, the simple integration lacks robustness for the multi-task nature of person search. This suggests the need for methods that enhance corruption robustness while considering the unique aspects of this

Table 2: **Individual evaluation of detection and reID stage** of end-to-end person search framework on CUHK-SYSU-C. 'Representation against Corruption' refers to the search performance when extracting person representation in corrupted images, assuming that detection was performed on clean images. 'Detection against Corruption' denotes the search performance when detection is conducted on corrupted images, coupled with the representations that are extracted from clean images.

| Severity | Representation against Corruption | | | | Detection against Corruption | | | |
| | Search | | Detection | | Search | | Detection | |
| | R@1 | mAP | Recall | AP | R@1 | mAP | Recall | AP |
|---|---|---|---|---|---|---|---|---|
| Oracle | 95.2 | 94.5 | 100.0 | 100.0 | 95.2 | 94.5 | 100.0 | 100.0 |
| Clean | 94.6 | 93.7 | 92.1 | 89.2 | 94.6 | 93.7 | 92.1 | 89.2 |
| Level 1 | 79.1 | 77.6 | 92.1 | 89.2 | 93.5 | 92.2 | 86.2 | 83.6 |
| Level 2 | 60.5 | 59.5 | 92.1 | 89.2 | 91.7 | 89.1 | 79.2 | 76.8 |
| Level 3 | 42.6 | 42.5 | 92.1 | 89.2 | 88.6 | 84.8 | 71.2 | 68.7 |
| Level 4 | 28.8 | 28.1 | 92.1 | 89.2 | 82.1 | 74.4 | 59.6 | 57.1 |
| Level 5 | 19.3 | 18.6 | 92.1 | 89.2 | 69.8 | 58.9 | 46.0 | 44.0 |

task. In the following sections, we explore the underlying reasons for corruption vulnerabilities in the person search for developing a method tailored to its unique challenges.

## 3.2 SENSITIVITY TO CORRUPTION IN DETECTION AND REPRESENTATION STAGES

The experiments in the previous section demonstrate that existing person search models are vulnerable to corruption. In a typical person search framework, the *detection* head first identifies person candidates, and then the reID head extracts *representations* from the detected regions. Therefore, we assess the individual sensitivity of both the detection and representation stages to corruption.

To evaluate detection sensitivity to corruption, we first apply the detection head to corrupted images to obtain predicted bounding boxes. We then extract features from the corresponding regions in clean images using these box coordinates and perform the search process. Conversely, to evaluate the representation sensitivity to corruption, we perform detection on clean images to obtain predicted boxes, then extract features from the corresponding regions in corrupted images, followed by the search process. We evaluate the model across five severity levels, providing a separate benchmark for each to observe how the model responds to increasing corruption severity.

The results of these individual sensitivity studies are in Table 2. The 'Oracle' row represents performance using ground truth bounding boxes, while the 'Clean' row refers to the results where both detection and search are performed on clean images. The five remaining rows (Level 1 through Level 5) show the impact of increasing the severity level of corruption. The results reveal that performance degrades more significantly when person representations are extracted from corrupted images. At level 5 of 'Detection against Corruption', although detection performance declines, search performance remains relatively stable, with 70% of R@1, indicating that the model retains the search capability despite noisy detection results. However, when person representations are corrupted (*i.e.*, 'Representation against Corruption'), search performance drops significantly, even with relatively strong detection performance. These findings reveal that both detection and representation stages are sensitive to corruption, further highlighting that the *representation stage is particularly vulnerable*.

## 3.3 INFLUENCE OF FOREGROUND & BACKGROUND FOR ROBUST PERSON REPRESENTATION

Unlike person re-identification, which relies on cropped images to extract person representations, person search processes entire scenes and thus uses larger receptive fields to extract representations. As a result, the background can significantly impact person search performance. To investigate the individual influence of corruption on the foreground and background, we selectively apply corruption to either region, as illustrated in Figure 4. We then measure search performance based on the representations obtained from the corrupted images across five severity levels.

Figure 4 presents search and detection performance under three corruption scenarios: corruption in the foreground ('fg corrupted'), corruption in the background ('bg corrupted'), and corruption on the entire scene ('both corrupted'). The results show that search performance remains largely unaffected when corruption is limited to the background. However, when corruption is applied to the foreground – directly impacting the person's appearance – there is a notable drop in both

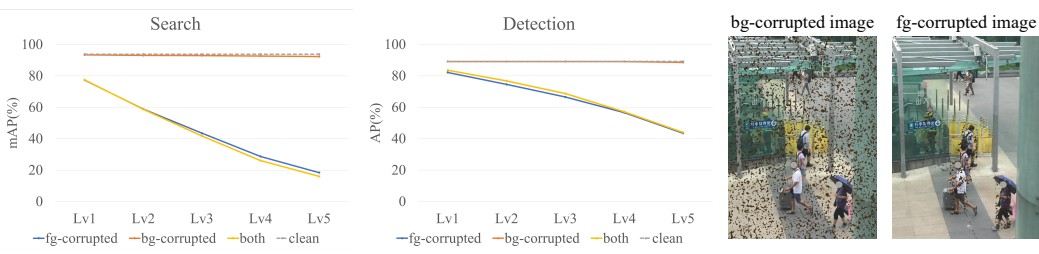

(a) Performance analysis by corruption region  (b) Corruption examples

Figure 4: **Performance evaluation on images from CUHK-SYSU where corruption is applied to either the background or the foreground.** 'bg-corrupted' indicates performance on images where corruption is applied only to the background, while 'fg-corrupted' refers to performance on images where corruption is applied only to the foreground (person) regions. 'both' represents performance on the CUHK-SYSU-C dataset, where corruption is applied to the entire image.

search and detection performance, comparable to the degradation observed when the entire scene is corrupted. This suggests that the integrity of the background plays a minor role in robust person representations, while *corruption on the foreground is much more detrimental*. Based on these observations, in the following section, we introduce a novel foreground augmentation approach with tailored regularization to achieve more robust person representations under corruption.

## 4 PROPOSED METHOD FOR ROBUST PERSON SEARCH TO CORRUPTION

### 4.1 FOREGROUND-AWARE AUGMENTATION

Data augmentation has proven to be a valuable approach for boosting model robustness in computer vision (Shorten & Khoshgoftaar, 2019; Rebuffi et al., 2021; Liu et al., 2024b). However, as shown in Figure 5 (b), applying data augmentation to the entire scene results in severe semantic corruption and unreliable bounding boxes (*e.g.*, overlapping pedestrians). Our analysis highlights that search performance remains largely unaffected when corruption is restricted to the background. To this end, we introduce a foreground-aware augmentation to generate appropriate augmented counterparts for given input images. We define the areas containing people as foreground and determine these using ground truth bounding boxes during training. To minimize the role of ground truth bounding boxes in generating shortcuts during training, we apply a translation to the box coordinates of a person before applying augmentation to the corresponding region.

In our training process, we incorporate both clean scenes and their augmented counterparts. Let the input image be denoted by $x^c$, and the set of people appearing in the scene be $P_{x^c} = \{p_1^c, \ldots, p_{n_x}^c\}$. Here, $n_x$ represents the total number of people appearing in the scene. We create $x^a$ by applying $\mathcal{T}$ to each person, where $\mathcal{T}$ represents the transformation by augmentation functions. The set of people appearing in $x^a$ is $P_{x^a} = \{p_1^a, p_2^a, \ldots, p_{n_{x^a}}^a\}$ and $p_j^a = \mathcal{T}(p_j^c)$. We employ Augmix (Hendrycks et al., 2020) and random erasing (Zhong et al., 2020) for our augmentation functions. For a rigorous robustness evaluation, we exclude augmentations that operate on similar principles to the corruptions used in creating CUHK-SYSU-C and PRW-C. With this in mind, the augmentations we use in our method are rotation, shearing, translation, solarization, autocontrast, equalization, and posterization. We use $x^c$ and $x^a$ to train the model with the OIM (Xiao et al., 2017) loss. Let $z_i \in \{z | z \in f(x^c; \theta) \cup f(x^a; \theta)\}$ be the normalized representation of a person $p_i$ from a scene, where $f(\cdot; \theta)$ is the function for extracting a set of person candidates on the given scene. Let $v_l$ be a vector in the lookup table where $l \in \{1, ..., L\}$, and $u_q$ be the $q$-th vector in the queue where $q \in \{1, ..., Q\}$. This lookup table serves as a memory bank containing representations for labeled persons, while the queue is a memory bank that stores representations for unlabeled persons. $L$ and $Q$ are the size of the lookup table and queue. The OIM loss is adopted as follows:

$$\mathcal{L}_{OIM} = \mathbb{E}_z[-\log h_i], \quad (1)$$

$$h_i = \frac{\exp(v_i^\top z_i / \tau)}{\sum_{l=1}^{L} \exp(v_l^\top z_i / \tau) + \sum_{q=1}^{Q} \exp(u_q^\top z_i / \tau)}, \quad (2)$$

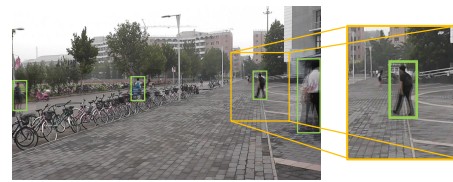 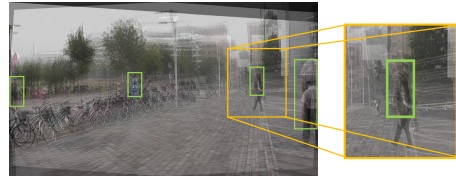

(a) Augmentation applied to foreground          (b) Augmentation applied to entire image

Figure 5: **Illustration of augmentation on foreground (a) and entire image (b)**. Unlike (a), naïve augmentation on the entire image can generate severe semantic perturbations. We define the foreground using ground-truth bounding boxes, and augmentation is applied to each area individually.

where $\tau$ is a temperature parameter. Each representation in the lookup table is updated with the momentum parameter $\eta$:

$$v_l \leftarrow \eta z_i + (1 - \eta)v_l. \tag{3}$$

## 4.2 REGULARIZATION FOR ROBUST PERSON REPRESENTATION

In Section 3.2, we observed that the representation stage was proven to be more susceptible to corruption than the detection stage. In this context, we propose a regularization method for person search to achieve robust person representation. Our goal is to achieve robust person representation through regularization by learning invariance between clean input images and their augmented versions. In person search frameworks, models typically generate multiple detected results for each person. Our regularization approach exploits this feature by utilizing these multiple detected results. To limit the excessive contribution of low-quality results, we utilize detected results where the Intersection over Union (IoU) with the ground truth bounding box is larger than $\alpha$. Let $h_j^c$ and $h_j^a$ be normalized representations obtained from the $j$-th person of original scene and its augmented counterpart. Using $h_j^c$, we define a mixture representation $M_j$ for the $j$-th person.

$$M_j = \sum_m s_{j,m} h_{j,m}^c, \tag{4}$$

For $h^a$ corresponding to detection results satisfying the constraint, we apply the Kullback-Leibler divergence. The regularization loss $L_{reg}$ is formulated as follows:

$$\mathcal{L}_{reg} = \frac{1}{n} \sum_j \sum_r s_{j,r} \text{KL}[h_{j,r}^a || M_j], \tag{5}$$

where

$$s_{j,*} = \frac{\exp(s_{j,*}/\tau_{iou})}{\sum_k^n \exp(s_{j,k}/\tau_{iou})}, \tag{6}$$

$s_{j,*}$ denotes the IoU score of $h_j^c$ or $h_j^a$ and $n$ indicates the number of detected results. Here, we detach $M_j$, aiming for invariant representation towards augmentation, and utilize the IoU scores $s_j$ as a weight, considering each target's quality.

Our proposed method can be seamlessly integrated into existing person search models. In the following section, we apply and validate our method to several state-of-the-art person search models.

## 5 EXPERIMENTS

**Implementation Details.** We use an ImageNet (Deng et al., 2009) pre-trained ResNet50 (He et al., 2016) as the backbone for all methods in our experiments. The queue size is set to 5000 and 500 for CUHK-SYSU and PRW, respectively. We use a gallery size of 100 when evaluating CUHK-SYSU-C. For PRW-C, the gallery size matches the total number of images in the test set. An initial learning rate of 0.003 is used. We employ SGD with a momentum of 0.9 and a weight decay of

Table 3: **Evaluation of the proposed method on various person search models** across clean and corrupted benchmarks.

| Method | CUHK-SYSU | | CUHK-SYSU-C | | | | PRW | | PRW-C | | | |
|---|---|---|---|---|---|---|---|---|---|---|---|---|
| | R@1 | mAP | rR@1 | rmAP | R@1 | mAP | R@1 | mAP | rR@1 | rmAP | R@1 | mAP |
| OIMNet (Xiao et al., 2017) | 87.7 | 86.2 | 37.7 | 36.7 | 33.0 | 31.7 | 76.7 | 37.3 | 45.8 | 23.6 | 35.2 | 8.8 |
| +Ours | **90.2** | **89.3** | **54.0** | **53.0** | **48.7** | **47.4** | **78.0** | **40.7** | **54.3** | **32.0** | **40.8** | **13.0** |
| NAE (Chen et al., 2020) | 92.3 | 91.4 | 44.3 | 42.4 | 40.9 | 38.8 | 80.6 | 42.8 | 42.6 | 20.9 | 34.3 | 8.9 |
| +Ours | **94.3** | **93.7** | **65.9** | **63.7** | **62.2** | **59.7** | **81.1** | **44.0** | **56.6** | **35.2** | **45.9** | **15.5** |
| OIMNet++ (Lee et al., 2022) | 94.0 | 93.2 | 38.6 | 37.0 | 36.3 | 34.5 | 83.2 | 47.3 | 43.9 | 19.9 | 36.5 | 9.4 |
| +Ours | **94.2** | **93.8** | **62.5** | **61.2** | **58.8** | **57.4** | **84.5** | **48.1** | **57.4** | **33.8** | **47.8** | **16.3** |
| SeqNet (Li & Miao, 2021) | 94.5 | 93.8 | 46.2 | 44.4 | 43.6 | 41.6 | 83.4 | 46.7 | 46.7 | 22.6 | 38.9 | 10.5 |
| +Ours | **94.9** | **94.3** | **70.3** | **68.9** | **66.7** | **65.0** | **84.0** | **47.0** | **57.0** | **33.8** | **47.8** | **15.9** |
| COAT (Yu et al., 2022) | **94.7** | **94.2** | 52.6 | 50.4 | 49.8 | 47.5 | **87.4** | **53.3** | 49.9 | 23.7 | 43.6 | 12.6 |
| +Ours | 92.8 | 92.1 | **68.9** | **67.4** | **63.9** | **62.0** | 86.9 | **53.3** | **55.4** | **40.1** | **55.4** | **21.4** |

Table 4: **Ablation study of our method** on CUHK-SYSU and CUHK-SYSU-C. Baseline refers to SeqNet (Li & Miao, 2021).

| Method | CUHK-SYSU | | CUHK-SYSU-C | |
|---|---|---|---|---|
| | R@1 | mAP | R@1 | mAP |
| Baseline | 94.5 | 93.8 | 43.6 | 41.6 |
| + Regularization | 94.2 | 93.4 | 55.6 | 53.7 |
| + Foreground-aware | 94.5 | 94.0 | 65.0 | 63.1 |
| + IoU Score | **94.9** | **94.3** | **66.7** | **65.0** |

Table 5: **Evaluation under real corruption scenarios.**

| Method | Dark | | Rain | |
|---|---|---|---|---|
| | R@1 | mAP | R@1 | mAP |
| Baseline | 33.3 | 34.7 | 31.6 | 35.9 |
| + Ours | **45.1** | **47.0** | **41.9** | **44.1** |

| Method | Blur | | Fog | |
|---|---|---|---|---|
| | R@1 | mAP | R@1 | mAP |
| Baseline | 64.0 | 64.0 | 56.8 | 47.8 |
| + Ours | **70.2** | **68.7** | **59.8** | **49.9** |

0.0005. For the hyper-parameters used in the training, the threshold value $\alpha$ for the IoU is set to 0.6. The momentum parameter $\eta$ is set to 0.5, and a temperature value $\tau$ is set to 0.2. The temperature parameter for the IoU score $\tau_{iou}$ is set to 0.6. We use an NVIDIA RTX 3090 GPU for the experiments.

**Results.** The experimental results are shown in Table 3. Our method shows promising performance improvements under corruption for all tested models. Specifically, the R@1 for NAE increases by 52% on CUHK-SYSU-C and 34% on PRW-C with our approach. It should be noted that the impact on clean dataset performance appears to be limited, with variations within ±5%. On the CUHK-SYSU dataset, our approach enhances the performance of all the original models except for COAT. COAT's method incorporates token mixup, which already offers inherent advantages in augmentation. These results suggest that our approach improves the robustness of person search models in corrupted environments without compromising their performance in clean conditions.

Note that our method performs better than the combination of well-known robustness methods in detection and re-identification (OADG+CIL) shown in Table 1. This suggests that our method is highly competitive with existing approaches to handling corruption.

**Ablation Study.** We conduct an ablation study on CUHK-SYSU and CUHK-SYSU-C to evaluate the effectiveness of each technique in Table 4, using SeqNet as our baseline. The second row, labeled 'Regularization', represents the method discussed in Section 4.2 that excludes the IoU score. The third row, labeled 'Foreground-aware', details changes in augmentation application from the entire scene to just the foreground area, while the fourth row, labeled 'IoU Score', shows results incorporating the IoU score.

Our findings indicate that all techniques—'Regularization', 'Foreground-aware' and 'IoU Score'—enhance search performance in corrupted environments, highlighting the effectiveness of our proposed method in tackling the challenges of person search under corrupted conditions.

**Validation of Proposed Method towards Real Corruptions.** To assess our method's effectiveness in real-world corruption cases, we collect images from BDD100K (Yu et al., 2020) and use them for our experiments. Since BDD100K is a comprehensive dataset acquired from tens of thousands of drivers across various locations, weather conditions, and time frames, it is commonly adopted to evaluate model robustness in corruption and associated research areas (Kim & Shin, 2024; Cygert

& Czyżewski, 2021; Kim & Shin, 2023). Although BDD100K is mainly used for detection and segmentation tasks, as it includes a diverse array of weather and temporal conditions, we gather scenes corresponding to several corruption scenarios from it and perform manual annotation. From this dataset, we gather scenes representing dark, rain, blur, and fog scenarios, then manually annotate the bounding boxes and identity labels of individuals present in these images. We adhere to the labeling protocol used in the creation of CUHK-SYSU. For evaluation, we use 100 images from each of the dark, rain, blur, and fog scenarios (400 total), following the default gallery size of the CUHK-SYSU evaluation protocol. Following the PRW evaluation protocol, we employ all images except the one containing the query person as the gallery and use all labeled persons as a query once. We train the models on the uncorrupted data (CUHK-SYSU) and evaluate their performances in each real corruption scenario. We make our annotations publicly available (annotation only), which include the information on the samples and label data we use (download link in the appendix).

Table 5 presents the model's performance evaluated across four real-world corruption scenarios. Our proposed method demonstrates performance improvements across all real corruption scenarios. Notably, in the dark corruption scenario, our method achieves a 12.3% mAP performance gain, showing the effectiveness of the proposed method leveraging our analyses. These results indicate that the proposed method can work in real-world corruption scenarios. Our method, which performs effectively under our proposed corruption, shows robustness in real-world corruption scenarios as well. See the appendix for a qualitative comparison of the proposed and real-world corruptions.

**Frequency Sensitivity Analysis.** To provide theoretical depth to our experiments, we analyze our proposed method from the perspective of frequency sensitivity. The 25x25 heatmap in Figure 6 represents the error rate (mAP) when evaluating the model using data corrupted with different frequency bases. The edges of the heatmap represent evaluations using data corrupted with high-frequency bases, while the center represents experiments performed with low-frequency bases. Red colors indicate higher error rates, while blue colors indicate lower error rates. Compared to the left heatmap (Baseline), the right heatmap (Ours) shows overall performance improvement across low, middle, and high frequencies. Yin et al. (2019) analyzed

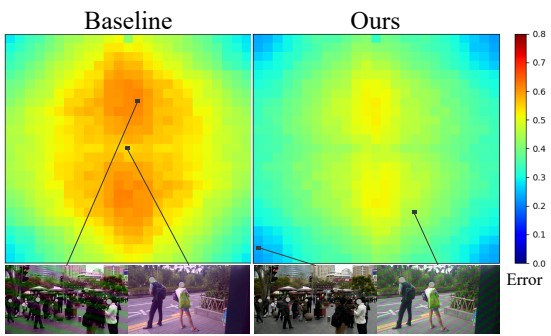

Figure 6: **Qualitative analysis of frequency sensitivity** between baseline (SeqNet) and our method.

the common corruptions in the frequency domain, showing that fog and contrast have relatively low-frequency components, noise-related corruptions have high-frequency components, and blur and pixelate have middle-frequency components. We think that our model's performance improvement on corruption benchmarks stems from its resilience to various frequency perturbations.

## 6 CONCLUSION

In this work, we present two benchmarks for evaluating the robustness of person search, CUHK-SYSU-C and PRW-C, to assess and analyze the robustness of person search models. We explore how various features of person search influence robustness with the expectation where our findings will be valuable lessons to the research community in related fields. From the findings obtained through our experiments, we propose a solution for robust person search that not only achieves competitive performance on clean datasets, but also demonstrates effective robustness enhancement in corrupted environments.

**Potential Limitations.** While our method performs well in simulated environments, we have not extensively tested its performance under real-world corruption scenarios. To mitigate this gap, we collect and evaluate our method on real corruption images, which shows the validity of our proposed approach in the real world. We have not considered scenarios with multiple simultaneous corruptions. Additionally, while we categorize corruption severity into five levels, this discretization may not fully capture the continuous nature of real-world corruption.

# Ethics Statement

−This section is not included in the page limit.−

## 7 Social Impact

The development and deployment of person search systems require careful consideration due to their potential impact on individual privacy and societal norms. While these systems offer substantial benefits for security and surveillance by enhancing the ability to locate and identify individuals across diverse environments, they simultaneously pose a profound risk to personal privacy. The ability to track and identify individuals without their consent can lead to a range of privacy violations, from the unwarranted monitoring of public movements to the potential for misuse in stalking and harassment. It is, therefore, imperative that developers, implementers, and policymakers involved in the creation and use of person search systems consider these ethical implications from the outset. The development of these systems should be guided by ethical principles that prioritize the well-being and privacy of individuals, incorporating mechanisms for accountability and oversight to prevent misuse. It is our collective responsibility to balance innovation with the imperative to safeguard human dignity and privacy.

By doing the above, we can harness the benefits of person search systems while mitigating the risks, ensuring these technologies enhance societal welfare without compromising individual freedoms. For instance, person search technology can expedite victim identification and rescue efforts in natural disasters or accidents, improving the effectiveness of emergency responses and potentially saving lives. Disaster environments often involve challenges like heavy rainfall or snowfall, while accident scenes may present issues such as varying lighting conditions or spatter-covered camera lenses. Our study aims to optimize person search techniques to function effectively under these varied and adverse conditions.

## 8 Further Discussions about Datasets

**Privacy.** Testing person search models in diverse situations and environments requires collecting new data for various situations with people in them, which can raise ethical issues. Since we provide 18 different scenarios for evaluating the person search models, our benchmarks serve as a good proxy for evaluating models across diverse environments without collecting new data. The parent datasets CUHK-SYSU (Xiao et al., 2017), PRW (Zheng et al., 2017) expose people's faces, and our datasets (CUHK-SYSU-C, PRW-C) inevitably inherit this issue. To mitigate the issue, we mask faces that are captured prominently and distinctly visible for the figures used in the paper to ensure their identities are not recognizable. Future researchers using the proposed CUHK-SYSU-C and PRW-C must be aware of this and take precautionary measures. When using them in research papers, we strongly recommend to de-identify individuals when their faces are clearly recognizable. Regarding the subset collected from BDD100K for Section 5, we ensure that the samples we utilize are from a data source that has already been publicly available, carefully adhering to its licensing terms.

**License.** We will release them in the form of code to generate our benchmarks from the parent datasets(CUHK-SYSU, PRW) rather than as raw images. The authors of the parent datasets provide guidelines for dataset usage on their respective websites, which are similar to the terms of CC BY-NC: For CUHK-SYSU, users are permitted to use the data only for non-commercial research and educational purposes. Users are not permitted to distribute the data. For PRW, they emphasize the purpose of non-commercial research applications. The dataset requires citation. For these reasons, we will also release CUHK-SYSU-C and PRW-C under CC BY-NC.

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

# Appendix

## A FURTHER DISCUSSIONS

### A.1 FURTHER DISCUSSION OF BENCHMARK DATASET DESIGN IN SECTION 3.1

Table 6 presents the case study that highlights a potential bias in person search models when the same type of corruption is applied to both query and gallery images. The table compares two scenarios: the case where corruption is applied only to the gallery images ('Gallery corrupted'), and the other case where the same corruption is applied to both query and gallery images ('Both corrupted'). Four different types of corruptions are examined: Brightness, Contrast, Saturate, and Spatter. Interestingly, for four corruption types, we observe higher performance when the same corruption is applied to both query and gallery images. This pattern suggests that when both query and gallery images are subjected to the same type of corruption, the model's ability to match them could improve. We consider this potential bias when designing our benchmark dataset, which is to randomly apply corruption types and severity levels to query and gallery images.

Table 6: **Case studies when the same corruption is applied to both query and gallery images**, it could lead to another bias, such that the similarity of the two images increases. CUHK-SYSU-C, severity level 5, SeqNet are used for the case studies, R@1 is used as an evaluation metric.

| Corruptions | Brightness | Contrast | Saturate | Spatter |
|---|---|---|---|---|
| Gallery corrupted | 71.5 | 11.7 | 55.9 | 60.9 |
| Both corrupted | 77.8 | 12.8 | 62.8 | 63.6 |

### A.2 MODEL EXPLANATION IN SECTION 3.1

In this section, we describe person search models that we examine in Section 3.1. These seminal state-of-the-art works have contributed to various aspects of the person search field. OIMNet (Xiao et al., 2017) firstly proposes an end-to-end person search framework by jointly training the detection and reID head. NAE (Chen et al., 2020) addresses the issue of conflicting learning objectives, a common challenge in person search and related fields (Xu et al., 2014; Zhang et al., 2021b; Lin et al., 2021), that arise during the joint learning of the detector and reID head. SeqNet (Li & Miao, 2021) improves the quality of detection results through a stronger detection head by considering that the detection result influences the training of the reID head. This simple yet effective concept has prompted subsequent studies to adopt its design (Li et al., 2022; Jaffe & Zakhor, 2023; Li et al., 2023). OIMNet++ (Lee et al., 2022) improves the widely used OIM (Xiao et al., 2017) loss and considers the quality of detection results in the training of the reID head. PSTR (Cao et al., 2022) and COAT (Yu et al., 2022) enable recent one-step person search frameworks to leverage the advantages of the Transformer (Vaswani, 2017).

### A.3 FURTHER DISCUSSION AND VALIDATION FOR EXPERIMENTS IN SECTION 3.2 AND SECTION 3.3

In Section 3.2 and 3.3 of the main paper, we analyzed which stage (representation or detection) is more susceptible to corruption and examined the influence of foreground and background regions on robust person representation. To validate these analyses and provide further insights, we conduct experiments that simultaneously examine both aspects. We analyze four scenarios: 'bg-corrupted & Representation against Corruption', 'bg-corrupted & Detection against Corruption', 'fg-corrupted & Representation against Corruption', and 'fg-corrupted & Detection against Corruption'. This experimental design allows us to examine the relative susceptibility of detection and representation stages under both background-only and foreground-only corruptions.

Table 7 presents our experimental results, revealing several key findings: In the case of 'fg-corrupted & Representation against Corruption', search performances align with the trends observed in both Figure 4's 'fg-corrupt' and Table 2's 'Representation against Corruption' cases; The performance from 'fg-corrupted & Detection against Corruption' shows similar the patterns seen in Figure 4's

Table 7: **Analysis of corruption effects on detection and representation stages under foreground and background corruptions.** The terms 'bg-corrupted', 'fg-corrupted', 'Representation against Corruption', and 'Detection against Corruption' have the same meanings as those used in Table 2 and Figure 4.

| | bg-corrupted | | | | | | | | fg-corrupted | | | | | | | |
|---|---|---|---|---|---|---|---|---|---|---|---|---|---|---|---|---|
| - | Representation against Corruption | | | | Detection against Corruption | | | | Representation against Corruption | | | | Detection against Corruption | | | |
| | Search | | Detection | | Search | | Detection | | Search | | Detection | | Search | | Detection | |
| severity | R@1 | mAP | recall | AP | R@1 | mAP | recall | AP | R@1 | mAP | recall | AP | R@1 | mAP | recall | AP |
| oracle | 95.2 | 94.5 | 100.0 | 100.0 | 95.2 | 94.5 | 100.0 | 100.0 | 95.2 | 94.5 | 100.0 | 100.0 | 95.2 | 94.5 | 100.0 | 100.0 |
| clean | 94.6 | 93.7 | 92.1 | 89.2 | 94.6 | 93.7 | 92.1 | 89.2 | 94.6 | 93.7 | 92.1 | 89.2 | 94.6 | 93.7 | 92.1 | 89.2 |
| level1 | 93.8 | 93.1 | 92.1 | 89.2 | 94.6 | 93.7 | 91.5 | 89.2 | 73.3 | 72.5 | 92.1 | 89.2 | 93.3 | 91.8 | 85.2 | 81.6 |
| level2 | 93.7 | 92.8 | 92.1 | 89.2 | 94.6 | 93.7 | 90.9 | 89.1 | 54.9 | 52.5 | 92.1 | 89.2 | 90.5 | 87.7 | 77.4 | 73.0 |
| level3 | 93.7 | 92.6 | 92.1 | 89.2 | 94.5 | 93.7 | 90.3 | 88.9 | 42.5 | 40.8 | 92.1 | 89.2 | 86.0 | 80.0 | 65.8 | 60.8 |
| level4 | 92.6 | 92.0 | 92.1 | 89.2 | 94.5 | 93.6 | 89.6 | 88.3 | 29.6 | 28.2 | 92.1 | 89.2 | 80.0 | 71.2 | 55.2 | 49.4 |
| level5 | 92.3 | 91.8 | 92.1 | 89.2 | 94.5 | 93.6 | 89.2 | 88.1 | 19.6 | 18.8 | 92.1 | 89.2 | 70.5 | 58.8 | 44.1 | 37.8 |

'fg-corrupt' and Table 2's 'Detection against Corrupt scenarios'; Both search and detection performances under background corruption maintain tendencies similar to those obtained with clean set. These results are consistent with previous findings, validating our observations.

# B    FURTHER ANALYSIS

## B.1    HYPERPARAMETER ANALYSIS

We conduct experiments to analyze four hyperparameters: the temperature for IoU score $\tau_{iou}$, types of augmentation used in $\mathcal{T}$, $\tau$, and the threshold $\alpha$. Table 8 shows the results of our analysis on the effect of $\tau_{iou}$. In this analysis, 'reverse' refers to the effect of our $\tau_{iou}$ applied in reverse. We implement this by applying $1 - \tau_{iou}$, which inverts the shape of the IoU distribution, before applying $\tau_{iou}$. The 'uniform' case represents an uniform IoU distribution, equivalent to not using the IoU score at all. Our results show that performance improves when using the IoU score (with $\tau_{iou}$ values of 0.4, 0.6, and 0.8) compared to the 'uniform' case where IoU is not used. Moreover, the 'uniform' case outperforms the 'reverse' case, which aligns with our intuition. The performance remains relatively stable across different hyperparameter values when using the IoU score.

Table 9 presents our analysis of the three hyperparameters: types of augmentation used in $\mathcal{T}$, $\tau$, and $\alpha$. For the analysis of augmentation $\mathcal{T}$, 'color' represents the evaluation result of a model trained using solarization, autocontrast, equalization, and posterization, while 'geometric' refers to the result of a model trained using rotation, shearing, and translation. The results indicate that the 'both' setting, which combines color and geometric augmentations, yields the best performance. This suggests that both types of augmentation contribute positively, with geometric augmentations showing more effectiveness. Regarding $\tau$ and $\alpha$, we set these parameters as values of 0.2 and 0.6, respectively, which shows the best performance.

Table 8: **Analysis for $\tau_{iou}$ impact.** The number indicates the value set for $\tau_{iou}$, *uniform* denotes not to apply the concept of IoU score, *reverse* indicates applying it in reverse.

| $\tau_{iou}$ | CUHK-SYSU-C | |
|---|---|---|
| | R@1 | mAP |
| *reverse* | 64.2 | 62.4 |
| *uniform* | 65.0 | 63.1 |
| 0.4 | 66.2 | 64.2 |
| 0.6 | **66.7** | **65.0** |
| 0.8 | 66.1 | 64.4 |

Table 9: **Analysis for impacts of $\mathcal{T}$, $\tau$, and $\alpha$.**

| Type | Settings | CUHK-SYSU-C R@1 | mAP | Type | Settings | CUHK-SYSU-C R@1 | mAP | Type | Settings | CUHK-SYSU-C R@1 | mAP |
|---|---|---|---|---|---|---|---|---|---|---|---|
| $\mathcal{T}$ | color | 54.5 | 52.3 | $\tau$ | 0.1 | 63.6 | 61.6 | $\alpha$ | 0.5 | 64.8 | 63.2 |
| | geometric | 57.5 | 55.3 | | 0.2 | **66.7** | **65.0** | | 0.6 | **66.7** | **65.0** |
| | both | **66.7** | **65.0** | | 0.3 | 63.2 | 62.0 | | 0.7 | 64.2 | 62.0 |

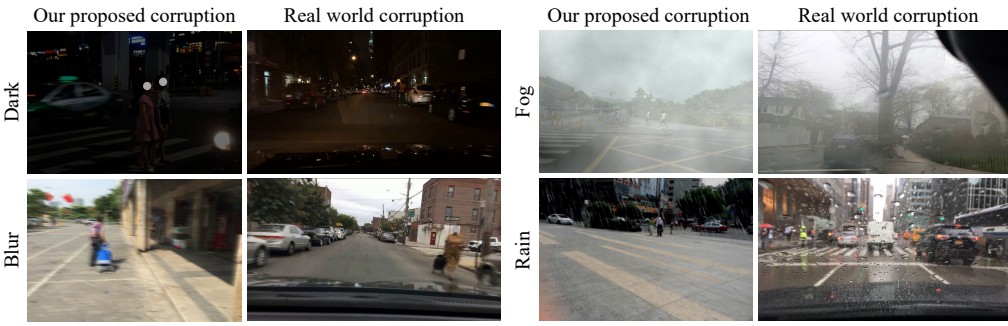

Figure 7: **Comparison of proposed corruptions with real-world corruption scenarios** for dark, blur, fog, and rain conditions.

## B.2 QUALITATIVE RESULTS

We present the qualitative results of evaluating different models on corruption scenarios. The results on PRW-C are shown in Figure 9. Each row represents the results of different models, and each column shows different query examples and the search results of each model accordingly. Blue indicates the location of the query person, red represents incorrect detection results by the model, and green represents correct detection results. By looking at examples 1 to 3, where our model successfully retrieves a person while other models fail, we can see the efficacy of our method in extracting robust representations of the person when corruption is applied. We also provide the result of the real-world corruption data (introduced in Section 5) in Figure 10.

## B.3 QUALITATIVE COMPARISON BETWEEN PROPOSED AND REAL-WORLD CORRUPTIONS

In this section, we compare our proposed corruptions with real-world corruptions gathered as described in Section 5. Figure 7 illustrates four corruption scenarios: dark, blur, rain, and fog. Our proposed corruption can capture distinct features in real-world corruption scenes. For example, the proposed dark reflects the reduced overall brightness observed in real-world blur scenes, the proposed blur presents decreased sharpness, and motion traces observed in real-world blur scenes. While our corruption benchmark assumes the presence of only one type of corruption in a scene, corruption in real-world images can be more complex. As shown in Figure 7, the real-world fog image contains some raindrops as well as a fog effect, while the real-world rain image exhibits blurriness. We aim to investigate these multiple corruption scenarios in our future research endeavors. The annotations we use for real-world corruption samples can be accessed in this link[1].

## B.4 PERFORMANCE COMPARISON ACROSS VARIOUS SEVERITY LEVELS

We conduct the experiment to evaluate the robustness of our proposed method across various severity levels when applied to five different person search models. The graph in Figure 8 illustrates the mAP performance of five person search models enhanced with our approach, compared against the OADG+CIL combination, across five levels of corruption severity. The graph shows a general downward trend in performance as severity increases. SeqNet+ours and COAT+ours show strong resilience, maintaining high mAP scores even at the most severe corruption levels. We observe

---

[1]Url: `https://drive.google.com/drive/folders/13z7nn9gesSTzXHKKSNk131zxvYMRTBoL?usp=drive_link`

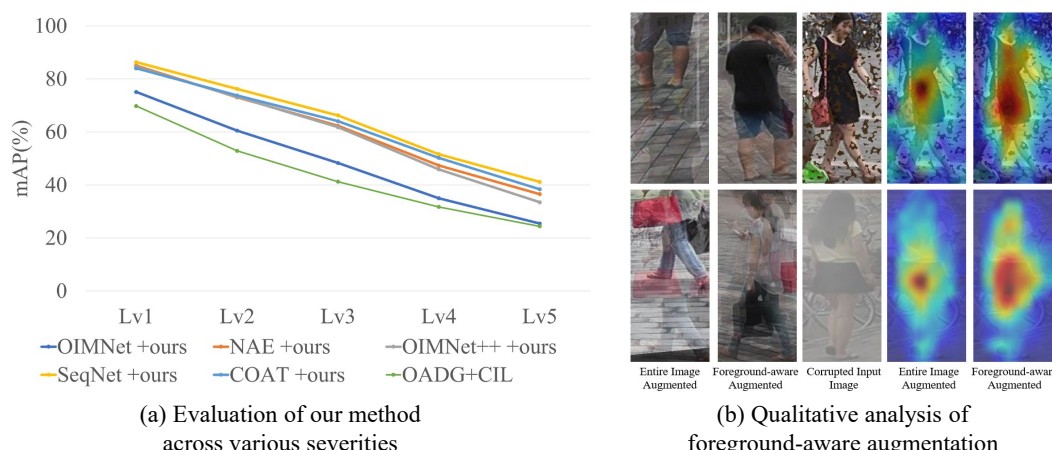

(a) Evaluation of our method
across various severities

(b) Qualitative analysis of
foreground-aware augmentation

Figure 8: **(a) Evaluation of our method across various severities.** We apply our method to existing five different person search models. OADG+CIL refers to the combination of existing works mentioned previously. **(b) Qualitative analysis of foreground-aware augmentation.** We compare the contents in bounding box regions after augmentation applied based on two regional criteria: entire image and foreground region. The last two columns denote the Grad-CAM analysis of two types of augmentations on two corruption scenarios (spatter, fog).

that even the basic OIMNet model, when enhanced with our method, consistently surpasses the performance of the OADG+CIL combination. This underscores the effectiveness of our approach, even when applied to simpler baseline models.

### B.5 FURTHER ANALYSIS ABOUT FOREGROUND-AWARE AUGMENTATION

To provide a deeper insight into our proposed foreground-aware augmentation, we present additional qualitative analysis. Figure 8 presents the results with and without foreground-aware strategy, along with corresponding Grad-CAM (Selvaraju et al., 2017) visualization results. The 'Entire Image Augmented' represents cases without foreground-aware strategy, while the 'Foreground-aware Augmented' shows cases with its application. The third, fourth, and fifth columns in each row show corrupted input images and their corresponding Grad-CAM results under two different corruptions (spatter, fog). We observe the bounding box regions cropped from the full images. In the first column, we can see that the naïve use could lead to the problem, such as severe semantic perturbation and unreliable bounding boxes. In contrast, the second column shows that our foreground-aware augmentation successfully applies augmentation while ensuring discriminative parts of the person remain within the bounding box. Accordingly, the results obtained from our strategy in the fifth column show that the model better captures the person's discriminative information.

## C BENCHMARK DETAILS

### C.1 BENCHMARK STATISTICS

Table 5 presents the statistics of CUHK-SYSU-C and PRW-C. The statistics regarding images, identities, and pedestrians are the same as those for the test split of CUHK-SYSU and PRW. As mentioned in the main paper, each image in CUHK-SYSU-C and PRW-C is randomly assigned to one of the 18 corruptions and randomly assigned to one of the five severity levels. We repeat this process several times (5) and report the average performance.

Table 10: **Statistics for CUHK-SYSU-C and PRW-C benchmarks.**

|  | # images | # identities | # pedestrians | # images per corruption | # images per severity |
|---|---|---|---|---|---|
| CUHK-SYSU-C | 6,978 | 2,900 | 40,871 | 388.6 | 1,395.6 |
| PRW-C | 6,112 | 544 | 25,062 | 339.5 | 1,222.4 |

## C.2 BIAS

CUHK-SYSU was collected from various locations in Chinese urban cities and movie scenes, while PRW was filmed at a Chinese university. Both datasets contain a sufficiently large number of individuals representing both genders. While PRW features a relatively younger demographic, CUHK-SYSU includes people of various ages. Both datasets have an ethnicity bias, predominantly featuring Asian individuals.

## C.3 CORRUPTION IMPLEMENTATION DETAILS

In creating the benchmarks for the person search, we consider the typical attributes of scenes used in person search, where both prominent subjects and multiple small background figures coexist in the same scene. We conduct the data quality check to ensure that the persons in the images are still detectable and re-identifiable by humans, even after corruption is applied. The implementation details of the corruption in CUHK-SYSU-C and PRW-C are as follows:

**Snow.** We use the method proposed in Hendrycks & Dietterich (2019) to implement the snow corruption. To represent the diverse features of the snowy scene in the real world, we express various attributes of snowy scenes and adjust their intensity for different severity levels. *Flake Size:* Determines the average size or thickness of the snowflakes. As this value increases, the size of the snowflakes increases. We set the parameters (0.1, 0.2, 0.55, 0.55, 0.55) to adjust its intensity. *Size Variation*: Represents the standard deviation of the size distribution of snowflakes. A larger value results in greater variation in the sizes of snowflakes. We set 0.3 as a parameter for the intensity. *Snowfall Intensity*: Indicates the degree of snowfall intensity applied to the image. Higher values simulate heavier snowfall. We set the parameters (3, 2, 4, 4.5, 2.5) to adjust its intensity. *Snow Coverage Threshold*: Sets the minimum value for snow generation. Snow will not be generated below this threshold, simulating areas with less snow coverage. We set the parameters (10, 12, 12, 12, 12) to adjust its intensity. *Wind Effect*: Determines the radius of motion blur. This simulates the direction and speed of falling snow, affected by wind. We set the parameters (0.5, 0.5, 0.9, 0.85, 0.85) to adjust its intensity. *Blurriness*: Determines the intensity of motion blur. Higher values make the snowflakes appear more blurred, simulating faster-falling snow or stronger wind. We set the parameters (4, 4, 8, 8, 12) to adjust its intensity. *Snow Opacity*: Determines the mixing ratio between the original image and the snow effect layer. Values closer to 1 show more of the original image, while values closer to 0 intensify the snow effect, simulating denser snowfall. We set the parameters (0.8, 0.7, 0.7, 0.65, 0.65) to adjust its intensity.

**Frost.** We use the method proposed in Hendrycks & Dietterich (2019) to implement the frost corruption. This simulates the effect of frost or ice forming on the surface of the image, giving the appearance of a cold and frosty environment. *Frost Intensity:* This determines the strength of the frost effect applied to the image. Higher values result in more pronounced frost, simulating thick ice or frost on the surface. We set the parameters (1, 0.8, 0.7, 0.65, 0.6) to adjust its intensity. *Blending Ratio:* Controls how much the frost image is blended with the original image. A lower value results in more frost coverage, while a higher value reveals more of the original image beneath the frost layer. We set the parameters (0.4, 0.6, 0.7, 0.7, 0.75) to adjust the blending ratio.

**Fog.** We use the method proposed in Hendrycks & Dietterich (2019) to implement the fog corruption. This simulates the effect of fog or mist, reducing visibility and softening the details in the image. *Fog Density:* This determines the density of the fog applied to the image. Higher values result in denser fog, which obscures more of the image. We set the parameters (1.5, 2.0, 2.5, 2.5, 3.0) to adjust the density. *Fog Smoothness:* Controls the rate of decay for the fractal noise used to generate the fog. Lower values create fog with sharper, more defined transitions, while higher values produce smoother fog with more gradual transitions. We set the parameters (2, 2, 1.7, 1.5, 1.4) to adjust its smoothness.

**Rain.** We use the method proposed in Saxena (2023) to implement the rain corruption. To simulate the rain effect on images, we introduce various attributes related to rain and adjust their intensity at different severity levels. *Slope of Rain Droplets:* Determines the inclination of rain droplets. This simulates how much the rain is tilted by the wind. At lower intensities, droplets are lighter and more

easily tilted by wind, forming steeper slopes. At higher intensities, droplets are heavier, causing rain to fall more vertically and be less affected by wind. We randomly select the slope from within the range (-20, 20) to adjust its intensity. ***Color of Rain Droplets:*** Determines how the rain droplets reflect light. As the severity increases, the color of the droplets changes from light gray to dark gray, simulating the visual effect of increasing rain density. This change reflects the reduction of light reflection and transmission through the air due to heavier rain. The default color is set to light gray (200, 200, 200), while for heavy rain, it is defined as medium gray (150, 150, 150), and for torrential rain, it is represented as dark gray (80, 80, 80). ***Blur Value:*** Indicates the degree of blur effect applied to the image. Higher values make the image blurry, simulating reduced visibility during heavy rainfall. We set the parameters (2, 3, 4, 5, 6) to adjust its intensity. ***Drop Length:*** Sets the length of rain droplets. At higher intensities, the length of droplets increases, creating a more dramatic rain effect. We set the parameters (10, 20, 30, 40, 50) to adjust its intensity. ***Drop Width:*** Determines the width of rain droplets. At higher intensities, the width of droplets increases, enhancing the more distinct rain effect. We set the parameters (1, 2, 3, 4, 5) to adjust its intensity. ***Brightness Adjustment:*** Adjusts the overall brightness of the image to simulate the dark environment of a rainy day. Lower values decrease the overall image brightness, creating a gloomy atmosphere. We set the parameters (0.7, 0.6, 0.5, 0.4, 0.3) to adjust its intensity.

**Dark.** We use the method proposed in Kong et al. (2024) to implement the dark corruption. To simulate darkness, we reduce the overall brightness of the image, making it appear darker. The lower the value, the darker the image becomes. We set the parameters (0.60, 0.54, 0.48, 0.42, 0.36) to adjust its intensity.

**Contrast.** We use the method proposed in Hendrycks & Dietterich (2019) to implement the contrast corruption. To simulate contrast, we adjust the difference in brightness between pixels based on the average brightness of the image. Lower values result in a reduction of contrast, causing the image to appear blurrier and colors to become more uniform. We set the parameters (0.4, 0.33, 0.26, 0.18, 0.1) to adjust its intensity.

**Gaussian Noise.** We use the method proposed in Hendrycks & Dietterich (2019) to implement the gaussian noise corruption. To simulate the gaussian noise, we adjust the noise intensity, which controls the standard deviation of the gaussian noise distribution applied to the image. The higher the value, the more pronounced the noise becomes. We set the parameters (0.05, 0.07, 0.09, 0.12, 0.15) to adjust its intensity.

**Speckle Noise.** We use the method proposed in Hendrycks & Dietterich (2019) to implement the speckle noise corruption. To simulate the speckle noise, we adjust the noise intensity that is multiplied by the pixel values themselves. The higher the value, the more the image is disrupted and appears to have a grainy mixture. We set the parameters (0.1, 0.2, 0.3, 0.4, 0.5) to adjust its intensity.

**Gaussian Blur.** We use the method proposed in Hendrycks & Dietterich (2019) to implement the gaussian blur corruption. To simulate the gaussian blur, we adjust the standard deviation of the Gaussian filter applied to the image. The higher the value, the more blurred the image appears. We set the parameters (1, 2, 3, 4, 5) to adjust its intensity.

**Motion Blur.** We use the method proposed in Hendrycks & Dietterich (2019) to implement the motion blur corruption. This simulates the blur caused by the movement of the camera or objects in the scene during exposure, producing a streaking effect. ***Motion Trace Length:*** This controls the length of the motion blur, representing how far objects have moved during the exposure. A higher radius results in a more pronounced blur effect, simulating faster movement. We set the parameters (10, 15, 15, 15, 20) to adjust its intensity. ***Blur Sharpness:*** Determines the sharpness of the motion blur. Higher values result in a smoother blur, while lower values retain more definition along the motion streak. We set the parameters (3, 5, 8, 12, 15) to adjust its sharpness. ***Angle Direction:*** Controls the angle at which the motion blur is applied, simulating motion in different directions. The angle is randomized within a certain range to simulate natural motion blur effects caused by different movements.

**Defocus Blur.** We use the method proposed in Hendrycks & Dietterich (2019) to implement the defocus blur corruption. To represent the diverse features of defocus blur in real-world photography, we express two attributes of defocus and adjust their intensity for different severity levels. ***Blur Kernel Size:*** Determines the size of the blur kernel. As the radius increases, the blurring spreads over a larger area, causing details to fade into the background. We set the parameters (3, 4, 5, 7, 9) to adjust its intensity. ***Blur Smoothness:*** Controls the smoothness or sharpness of the blur effect. Higher values produce a smoother, more gradual blur, while lower values retain sharper transitions at the edges of the blur. This affects the overall softness of the defocus effect. We set the parameters (0.1, 0.5, 0.5, 0.5, 0.5) to adjust its intensity.

**Glass Blur.** We use the method proposed in Hendrycks & Dietterich (2019) to implement the glass blur corruption. To represent the diverse features of the glass blur scene in the real world, we express various attributes of glass blur and adjust their intensity for different severity levels. ***Blur Strength:*** Determines the strength of the glass blur applied to the image, simulating distortion as seen through the glass. Higher values result in a more blurred and diffused image, where details become softer and less defined. As sigma increases, the overall smoothness of the blur effect intensifies. We set the parameters (0.7, 0.9, 1, 1.1, 1.5) to adjust its intensity. ***Glass Distortion Magnitude:*** Represents the degree of distortion caused by imperfections in the glass. As this value increases, the image appears more warped, simulating the effect of viewing through glass with varying thickness or composition. We set the parameters (1, 2, 2, 3, 4) to adjust its intensity. ***Distortion Repetitions (Iterations):*** Determines how many times the displacement effect is applied, creating multiple layers of distortion. More iterations result in a more pronounced and complex glass-like effect. We set the parameters (2, 1, 3, 2, 2) to adjust its intensity.

**Elastic Transform.** We use the method proposed in Hendrycks & Dietterich (2019) to implement the elastic transform corruption. To simulate elastic distortion, we apply random, small-scale deformations to the image pixels, adding a flexible, rubber-like warping effect. The higher the value, the more pronounced the distortions become, making the image appear more heavily warped. We set the parameters (12.5, 16.25, 21.25, 25, 30) to adjust its intensity.

**Spatter.** We use the method proposed in Hendrycks & Dietterich (2019) to implement the spatter corruption. This method simulates liquid splashes or mud spatters on the image, which can occur in outdoor or dirty environments, distorting visibility and adding a natural effect of environmental interference. ***Liquid Amount:*** This determines the average amount of liquid spattered on the image. Higher values simulate heavier splashes or more liquid, resulting in larger and more widespread spatter areas. We set the parameters (0.65, 0.65, 0.65, 0.65, 0.67) to adjust its intensity. ***Size Variation:*** Represents the standard deviation of the size of spatter drops. A larger value results in more variation in the size of splatter particles, simulating irregular drops. We set the parameters (0.3, 0.3, 0.3, 0.3, 0.4) to adjust its intensity. ***Blur Radius:*** Determines the size of the gaussian blur applied to the liquid layer. Higher values create more diffused spatter, simulating less defined edges and softer splashes. We set the parameters (4, 3, 2, 1, 1) to adjust the blurriness. ***Coverage Threshold:*** This sets the minimum intensity threshold for spatter formation. Spatter below this threshold is not visible, simulating splatters that did not adhere to the surface. We set the parameters (0.69, 0.68, 0.68, 0.65, 0.65) to adjust its coverage. ***Opacity:*** This controls the opacity of the spatter layer. Higher values create more opaque spatter, making it more prominent on the image. Lower values create more transparent splatter effects, simulating thinner layers of liquid. We set the parameters (0.6, 0.6, 0.5, 1.5, 1.5) to adjust its visibility. ***Mud vs. Water Effect:*** This determines whether the spatter simulates water (0) or mud (1). When set to 1, the spatter is brown and more opaque, simulating thick mud. When set to 0, the spatter simulates pale water splashes. We set the parameters (0, 0, 0, 1, 1) to adjust its effect.

**Saturate.** We use the method proposed in Hendrycks & Dietterich (2019) to implement the saturate corruption. To represent varying levels of color saturation in real-world scenarios, we express different attributes that influence color intensity and adjust them for different severity levels. ***Saturation Scale:*** This determines the strength of the color saturation applied to the image. Higher values result in more vivid and intense colors, while lower values make the image appear more desaturated or washed out. Both high or low saturation scales can lead to distortion and degradation of the clean image. We set the parameters (2, 0.2, 0.1, 0.3, 5) to adjust its intensity. ***Offset:*** Represents a con-

stant value added to the saturation scale. It controls the base level of saturation applied uniformly across the image, ensuring that even low saturation images retain some color vibrance. We set the parameter (0, 0, 0, 0, 0.1) to adjust the base intensity.

**Pixelate.** We use the method proposed in Hendrycks & Dietterich (2019) to implement the pixelate corruption. To simulate pixelation, we reduce the resolution of the image, converting it into larger blocks of pixels, then resizing it back to the original resolution, removing details. The lower the value, the stronger the pixelation effect, making the image appear more blocky. We set the parameters (0.6, 0.5, 0.4, 0.3, 0.25) to adjust its intensity.

**JPEG Compression.** We use the method proposed in Hendrycks & Dietterich (2019) to implement the JPEG compression corruption. To simulate JPEG compression, we adjust the compression quality level to observe its effects on image quality. The lower the value, the more the image quality degrades, leading to more artifacts and damage. We set the parameters (25, 18, 15, 10, 7) to adjust its intensity.

**Brightness.** We use the method proposed in Hendrycks & Dietterich (2019) to implement the brightness corruption. To simulate brightness, we adjust the overall brightness of the image by adding a constant value to the pixel values. The higher the value, the brighter the image becomes, and the lower the value, the darker the image appears. We set the parameters (0.1, 0.2, 0.3, 0.4, 0.5) to adjust its intensity.

Table 11: **Performance by different types of corruptions.**

| Type | Snow | Frost | Fog | Rain | Dark | Contrast |
|---|---|---|---|---|---|---|
| R@1/mAP | 59.9 / 55.5 | 64.6 / 63.1 | 84.8 / 83.3 | 72.5 / 70.0 | 83.5 / 82.3 | 62.9 / 58.5 |

| Type | Gaussian Noise | Speckle Noise | Gaussian Blur | Motion Blur | Defocus Blur | Glass Blur |
|---|---|---|---|---|---|---|
| R@1/mAP | 32.3 / 27.8 | 71.0 / 68.9 | 79.2 / 75.6 | 79.1 / 74.5 | 79.3 / 76.5 | 81.5 / 79.0 |

| Type | Elastic Transform | Spatter | Saturate | Pixelate | JPEG Compression | Brightness |
|---|---|---|---|---|---|---|
| R@1/mAP | 92.6 / 91.7 | 79.5 / 77.7 | 54.2 / 45.0 | 78.8 / 76.7 | 78.3 / 73.2 | 90.3 / 89.3 |

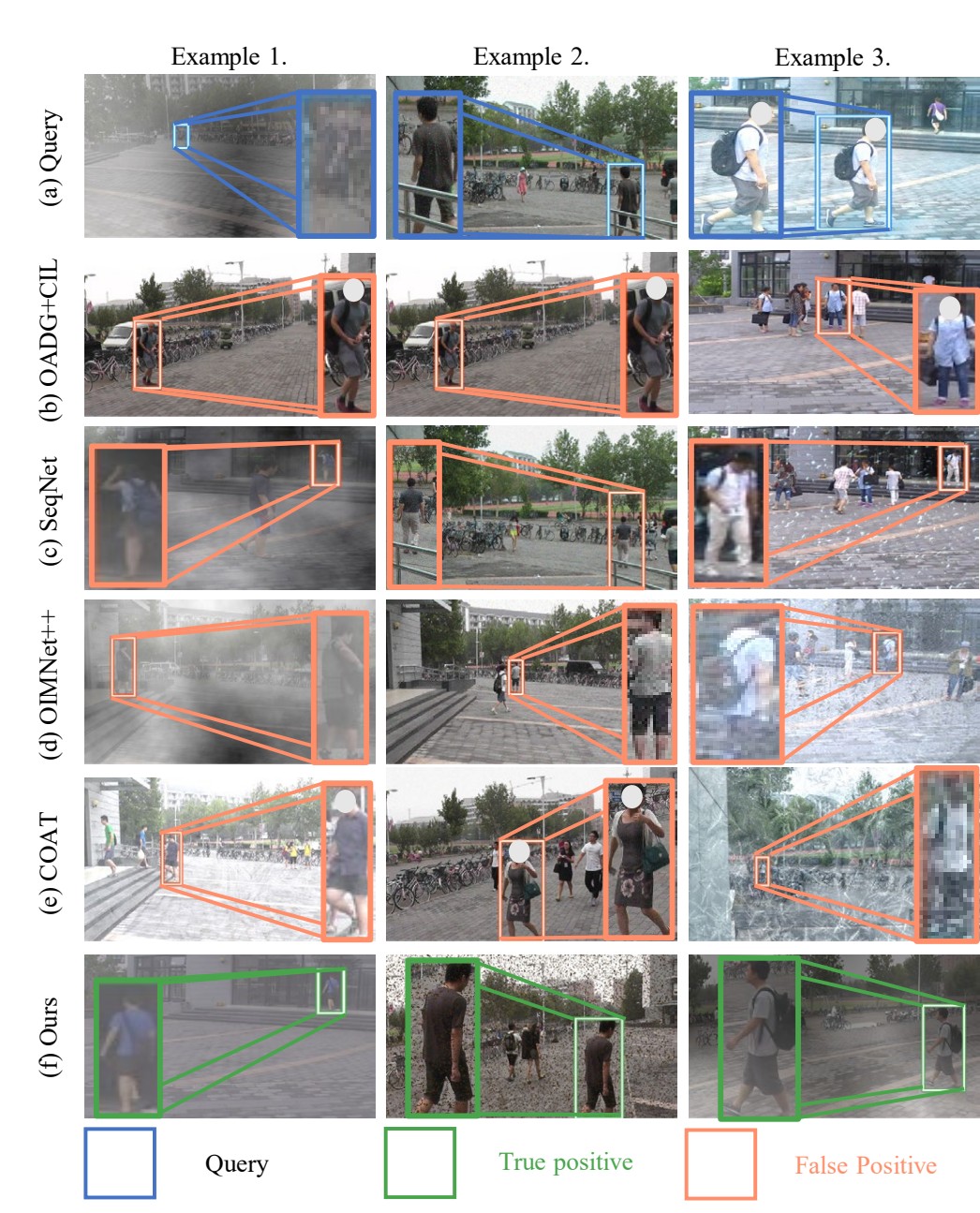

Figure 9: **Qualitative results of person search** on the PRW-C Dataset. The first row displays query images, while the second, third, fourth, and fifth rows show the results from OADG+CIL (Lee et al., 2024; Chen et al., 2021), SeqNet (Li & Miao, 2021), OIMNet++ (Lee et al., 2022), COAT (Yu et al., 2022), and Ours with SeqNet, respectively. Each column indicates the different query and the corresponding retrieval results of various models. The blue color denotes the box for a query, the red color indicates the box for failure cases, and the green color represents the box for success cases. A total of 18 types of corruption and 5 levels of severity are involved in establishing the PRW-C dataset.

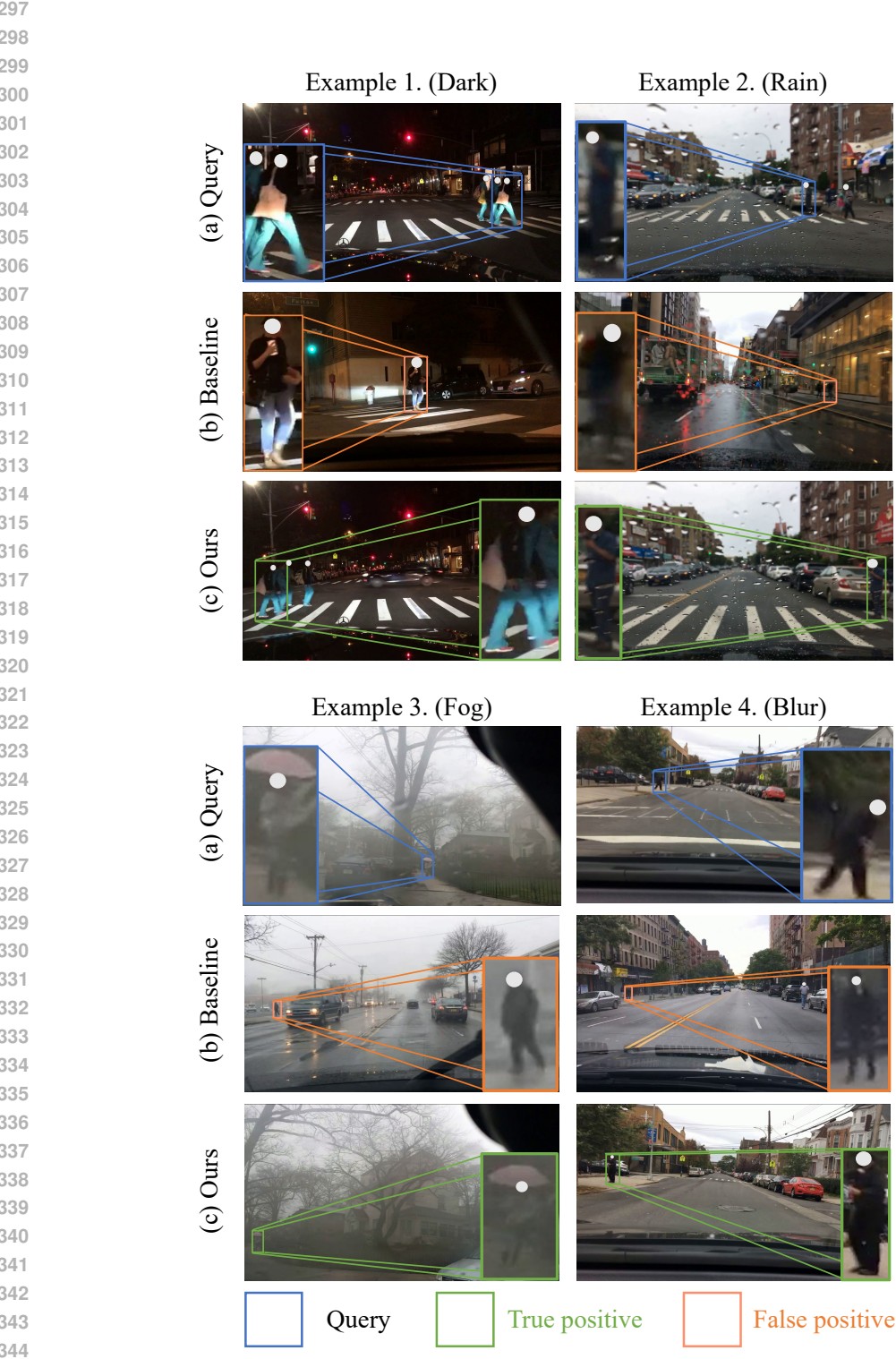

Figure 10: **Qualitative results of baseline and our method on real-world corruptions.** Baseline refers to SeqNet (Li & Miao, 2021).

