# OpenReview forum: "Towards Robustness of Person Search against Corruptions"
_ICLR.cc/2025/Conference — Submitted to ICLR 2025_

### Official Review · Reviewer_ujfi · 2024-10-31

**Soundness:** 3
**Presentation:** 4
**Contribution:** 4
**Rating:** 8
**Confidence:** 4

**Summary:**

This paper addresses the robustness of person search under various corruption scenarios. The authors introduce two new benchmarks, CUHK-SYSU-C and PRW-C, to evaluate person search models' robustness. Unlike previous studies that focused on independent tasks such as re-identification and detection, this work highlights the inadequacy of merely combining robust detection and re-identification models for robust person search. The authors investigate the sensitivity of detection and representation stages to corruption and its impact on background regions. Based on these insights, they propose a method that significantly enhances the robustness of existing person search models.

**Strengths:**

Overall, this paper takes a practical approach, considering numerous details and unexplored aspects of previous work. It introduces two benchmark datasets and evaluates leading models from both detection and re-identification perspectives. By analyzing the characteristics of foreground and background in the constructed datasets, the authors propose corresponding modules and validate the model's effectiveness.

1.Originality: This paper identifies shortcomings in previous datasets and evaluation methods, proposes a novel data construction approach, and discovers the varying impact of foreground and background on person search. These contributions demonstrate a certain level of originality.

2.Quality: The paper is thorough and provides valuable insights. The introduction and validation of large-scale datasets are of high quality.

3.Clarity: The paper is easy to follow. The problem statement, in particular, is logically clear and easy to understand.

4.Significance: The proposed datasets will assist future researchers in developing more robust end-to-end person search models. Additionally, the identified interference factors and their impacts will guide improvements in detection, re-identification, and other related fields.

**Weaknesses:**

This article has no obvious shortcomings. The author has done a large amount of related work and relevant verification. In my opinion, the workload of this article is sufficient. It also presents visualization analysis in various aspects and in the frequency domain, which is helpful for promoting research in this field.

**Questions:**

1.The author can consider expanding the dataset to the multi-modal re-identification area. Because the corruption types such as darkness and strong contrast constructed in the article are similar to the harsh visual environment in RGBNT201 for the destruction of visible light images. If the corruption can be extended to near-infrared or infrared images, or if LLM is used for text annotation of images and corruption is performed on the text, these are all good expansion ideas.

---

> ### Author Response · Authors · 2024-11-21
>
> Dear Reviewer ujfi,
>
> We sincerely appreciate your insightful suggestion and are grateful for your positive remarks on the motivation and insights behind our paper.
>
> The areas you mentioned are indeed crucial topics actively being researched in the fields.
> Given that collecting data in diverse and specific scenarios is costly and could raise ethical concerns, our benchmarks can serve as a scalable proxy for harsh environments across various situations.
> We agree that infrared and near-infrared scenarios represent practically important challenges in person retrieval applications, extending these corruption scenarios will provide significant value.
>
> Recent studies have shown that well-designed multi-modal approaches show higher generalizability compared to single-modal methods [1,2,3], and exploring this topic in person retrieval research is a good extension option.
>
> With the rising popularity of LLMs, obtaining rich textual annotations for images has become more feasible, computer vision fields are increasingly embracing multi-modal paradigms.
> Each modality can face different distinct corruption patterns, we believe robustness analysis in multi-modal scenarios provides valuable insights for this field as well.
> We are grateful for your insightful feedback, we will not stop here and are eager to explore these extensions.
>
> ---
>
> References:
>
> [1] CLIP: Learning Transferable Visual Models From Natural Language Supervision (ICML 21)
>
> [2] BLIP: Bootstrapping Language-Image Pre-training for Unified Vision-Language Understanding and Generation (ICML 22)
>
> [3] PLIP: Language-Image Pre-training for Person Representation Learning (NeurIPS 24)

---

> > ### Comment · Reviewer_ujfi · 2024-11-26
> > **Thank you for your response. I believe this paper meets the standards for publication at ICLR.**
> >
> > In future work, efforts could be made to extend this research to Multi-modal ReID, providing a comprehensive evaluation benchmark for multi-source data fusion. This would further enhance the impact of the field, particularly with datasets such as RGBNT201, RGBNT100, and MSVR310. We look forward to your continued contributions in this direction.

---

> ### Comment · Area_Chair_3nhm · 2024-11-27
>
> Dear Reviewer ujfi,
>
> In general, full-mark paper is likely to address a highly significant issue within a particular field, offering high-quality research and substantial contributions. This paper receives divergent reviews. Absolutely, other reviewers have provided constructive comments for the limitations and weakness of this paper. After carefully going through this paper, I believe the presentation and writing should be improved. Could you share your thoughts on why you believe this paper deserves a perfect score? Otherwise, it will receive an unethical flag for your review.

---

> ### Comment · Reviewer_ujfi · 2024-11-27
> **My thoughts about this paper.**
>
> Thank you for your reminder. In fact, the score of 10 that I gave was based on two main reasons:
> 1. I had previously reviewed this paper for NeurIPS. In my last review, I raised several questions, and this time, upon receiving the paper again, I carefully examined the detailed content. I found that the core issues I raised were thoroughly addressed in the main paper, particularly in the final two pages, including the frequency analysis visualizations and the impact of foreground and background. As a result, I gave the paper a high score of 8 during the initial review.
> 2. Due to my limited reviewing experience, I was not entirely clear about the grading scale for a score of 10. Based on your feedback, I am willing to adjust the score to 8 as you suggested.

---

### Official Review · Reviewer_Yohg · 2024-11-02

**Soundness:** 3
**Presentation:** 3
**Contribution:** 2
**Rating:** 3
**Confidence:** 4

**Summary:**

This paper introduces two benchmarks, CUHK-SYSU-C and PRW-C, evaluating the robustness of person search models under corruption scenarios. The authors propose a foreground augmentation and regularization method, improving the robustness.

**Strengths:**

The author examines the shortcomings of existing ReID methods based on experiments, explaining the issues in a clear and easy-to-understand manner. The authors use extensive experiments to demonstrate the drawback of existing methods.

**Weaknesses:**

1.	The five severity levels of the proposed benchmark are not detailed in either the main text or Fig. 1, which is quite important.
2.	The benchmark adds noise to the test sets of existing datasets without collecting additional person images for specific scenarios. I believe this does not qualify as a benchmark that makes a significant contribution. It it better to be considered as an evaluation metric.
3.	The proposed module is an existing augmentation, which I believe is quite incremental in terms of novelty.
4.	CUHK-SYSU and PRW datasets are relatively small, the proposed method has not been validated on larger-scale datasets. Therefore, the scalability of the method is doubted.
5.	Although the author claims that the augmentation method is different from the construction of the benchmark, they use a data augmentation method to solve an augmented test set of existing dataset. This is intuitive and within expectation, and I did not find anything particularly innovative in this approach.
6.	Several related works are missing, including but not limited to “SAT: Scale-Augmented Transformer for Person Search”, and “Making person search enjoy the merits of person re-identification”. Especially, the method of SAT incorporates the augmentation into transformer, whose contribution if believed to be of value.
7.	There are a lot typos in the manuscript, such as the symbols for the loss functions are inconsistent in Eq 5.

**Questions:**

1.	What is the five severity levels of the proposed benchmark, or can you quantatitively describe the differences between them?
2.	It is better to compare more baselines other than the five baselines in Tab. 3.

**Details Of Ethics Concerns:**

Containing person images, which involves privacy.

---

> ### Author Response · Authors · 2024-11-23
> **Official Comment by Authors (1/2)**
>
> Dear Reviewer Yohg,
>
> Thank you for constructive reviews and your time and effort in evaluating our work. Below, we address your concerns and questions.
>
> ---
>
> ### **About missing severity level description in the paper**
>
> We would like to refer to Lines 200-205 in Section 3.1 of the main paper, which provides an explanation of severity levels using rain corruption as a representative example.
>
> Appendix C.3 contains comprehensive implementation details for all 18 corruption types, including control factors and their quantitative explanation of all parameter values constructing our framework.
>
> From Appendix, we bring the description of how we define the severity using snow as an example. We adjust several attributes:
>
> - Snow density: The heavier the snow, the more snowfall appears.
>
> - Snow particle size: As the snow becomes heavier, the size of the snowflakes increases.
>
> - Snow particle transparency: As the snowfall becomes heavier, there's typically more moisture in the air, leading to a higher probability of water droplets combining to form larger and more distinct snowflakes.
>
> - Brightness of scene: Snow increases the brightness of darker areas in the image, simulating how snow reflects light.
>
> - Slope of snow: This simulates how much the snow is tilted by the wind. At lower severity levels, the snowflakes are lighter in weight and thus more easily tilted by the wind, resulting in a steeper slope. At higher severity levels, the snow falls more vertically to the ground, as heavier snowflakes are less affected by wind.
>
> We define the severity for person search task through data quality check, to ensure persons in the images are still detectable and re-identifiable by humans, even at the highest severity.
>
> ---
>
> ### **About adding more related works**
>
> Thank you for your suggestion.
>
> We conduct the experiment of Tab 3, with ***three additional related works*** [9,10,11], while we were unable to include the other suggested model in our current rebuttal timeline due to code availability issue. We evaluate the model on the CUHK-SYSU and PRW, we validate our approach on this model as well.
>
> We appreciate your feedback and will include the experiment results of [9,10,11] in our revision.
>
> |Method|CUHK|CUHK-C|CUHK-C|PRW|PRW-C|PRW-C|
> |-|-|-|-|-|-|-|
> ||R@1/mAP|rR@1/rmAP|R@1/mAP|R@1/mAP|rR@1/rmAP|R@1/mAP|
> |PS-ARM [9]|94.8/94.1|50.2/48.5|47.5/45.6|85.2/52.0|52.4/27.2|44.7/14.1|
> |+Ours|94.0/93.7|70.3/68.2|66.1/63.9|84.7/50.2|61.1/38.1|51.8/19.1|
> |HKD [10]|94.9/94.2|48.8/47.0|46.4/44.3|85.1/51.5|49.8/24.4|42.4/12.5|
> |+Ours|94.9/94.4|70.0/68.2|66.4/64.3|84.2/50.0|58.0/36.8|48.8/18.4|
> |SAT [11]|94.8/94.4|47.3/45.3|44.8/42.7|87.5/54.5|51.1/24.6|44.7/13.4|
> |+Ours|94.2/94.0|70.4/68.3|66.3/64.2|86.1/53.4|65.3/40.9|56.2/21.8|
>
> ---
>
> ### **About the scalability of the method**
> As CUHK-SYSU and PRW are the two major datasets for person search, and the models we use are evaluated and validated on these two benchmarks, we use them to allow for comparison with existing evaluations on the same basis.
>
> Our evaluation framework for corruption robustness is designed to be seamlessly adaptive to existing dataset.
>
> To further validate our method's scalability, we conducted additional experiments on PoseTrack21[7], a large-scale dataset recently adopted in person search research [8]. The experimental results show that our method performs effectively on large-scale dataset as well.
>
> |Method|PoseTrack21-C|PoseTrack21-C|PoseTrack21|
> |---|---|---|---|
> ||rR@1/rmAP|R@1/mAP|R@1/mAP|
> |OIMNet++|71.3/27.4|63.3/16.6|88.7/60.6|
> |+Ours|78.4/39.9|70.7/24.5|90.1/61.4|
> |SeqNet|69.6/28.7|59.9/17.0|86.1/59.2|
> |+Ours|81.0/46.0|70.0/27.5|86.4/59.7|
>
> ---
>
> ### **About methodological innovation**
>
> Even though we leverage some existing techniques such as AugMix, we have designed our method tailored to the person search task, based on our experimental observations.
>
> Our method, though relatively simple in its design, derives from our robustness analysis and observations of the person search framework, showing to be simple-yet-effective in its performance. To the best of our knowledge, we are the first to introduce selective augmentation and its corresponding regularization approach in the person search field.
>
> Also, its simplicity enables its broad applicability across various person search models without incurring additional parameter or inference time costs.
>
> With the advantage that our method maintains clean set performance within a ±2% range, it pushes the robustness performance of diverse state-of-the-art models to a large margin, where models themselves show a limited resilience to corruptions in Tab 1.
>
> Nonetheless, we also agree that future technique-focused research, such as advanced architectures for robust person search, can also be conducted, which we believe is a promising future direction. We believe our comprehensive framework and analyses will serve as a valuable foundation for such future technical innovations in the field.

---

> ### Author Response · Authors · 2024-11-23
> **Official Comment by Authors (2/2)**
>
> ### **About the value of our benchmark beyond evaluation metrics**
>
> In the field of corruption robustness, the prevalent approach is to evaluate the model performance by adding noise to the image, which has contributed to establish the framework for robustness analysis across various computer vision fields (Yi et al. 2021; Chen et al. 2021; Kong et al. 2023), and it encourages continuous exploration in this direction [1,2 Lee et al. 2022].
>
> Although our framework leverages some existing techniques, ***its design is specialized with task-specific insights for evaluating person search framework***, beyond merely employing the existing approach.
>
> - **Our framework construction is guided by experimental observation.** Based on the findings in Appendix A.1, we apply different corruptions to query and gallery images. As demonstrated in Appendix Table 6, applying identical corruptions to both query and gallery images could introduce bias by artificially increasing image similarity. This design also reflects the insight that query and gallery images may not always exhibit the same type or severity of corruption in real-world.
>
> - **We carefully select corruption types and calibrate their severity levels optimized for evaluating person search framework.** In deciding types of corruption, we focus on real-world deployment scenarios while maintaining comprehensive coverage of corruption types. Given that noise-related corruptions in ImageNet-C have an overlapping score close to 1 [3], we replace shot-noise and impulse-noise with rain and dark, which better represent realistic deployment scenarios. We calibrate severity levels to be challenging while ensuring that persons in the images remain detectable and re-identifiable by humans even at the highest severity.
>
> - **Mitigating ethical concerns of new data collection**: As the person search field demands careful ethical consideration due to its use of human data, our methodology provides an analysis framework containing diverse corruption scenarios without ethical concerns.
>
> We would like to emphasize that even though we leverage the existing technique, our framework derive the first pioneering analysis on this topic, thus providing a comprehensive framework where the evaluations, analyses, and further methodological research can be encouraged.
>
> |Real-world Validation|Fog|Dark|Rain|Blur|
> |-|-|-|-|-|
> |**Training data \ Metric**|R@1/mAP|R@1/mAP|R@1/mAP|R@1/mAP|
> |CUHK-SYSU|56.8/47.8|33.3/34.7|31.6/35.9|64.0/64.0|
> |CUHK-SYSU-C|**65.0/55.9**|**40.3/43.2**|**37.7/42.7**|**67.8/67.1**|
>
> We conduct experiments to further validate the value of our corruption benchmark.
> Instead of using clean data for training, we use our corruption benchmark for training and test the performance on 4 corruption scenarios -fog, dark, rain and blur- in BDD100K, a real-world corruption introduced in Tab 5. The results show training with our corruption benchmark helps improve the performance, ***exhibiting the value that our benchmark can bridge the distribution gap towards real world***.
>
> ---
>
> ### **About the difference between augmentation and corruption**
>
> Data augmentation is a popular approach for enhancing model robustness and generalizability.
>
> Data augmentation has been widely utilized in domain generalization[4,5], and it has been revealed by the survey that data augmentation is the most widely adopted[6], which lead us to pioneer the application of data augmentation for robustness in person search.
>
> Distinct from general adoptions, leveraging our observations, we tailor the usage of the augmentation techniques to the person search framework by selectively applying them to the foreground regions.
>
> Furthermore, our method shows the effectiveness not only on our corruptions, but also on the real corruption scenarios, as validated in Tab 5 of main paper.
>
> ---
>
> ### **About notations**
> We have fixed some typos in our submission. To clarify potential ambiguity in Eq 5, we have revised the notation $h$ to $h_c$.
>
> We welcome the opportunity to provide further clarification if any points remain unclear.
>
> ---
>
> [1] Robust Heterogeneous Federated Learning under Data Corruption (ICCV 23)
>
> [2] Towards Better Robustness against Common Corruptions for Unsupervised Domain Adaptation (ICCV 23)
>
> [3] Using the Overlapping Score to Improve Corruption Benchmarks (ICIP 21)
>
> [4] Generalizing to Unseen Domains: A Survey on Domain Generalization (IJCAI 21)
>
> [5] Domain Generalization: A Survey (TPAMI 23)
>
> [6] A Survey on the Robustness of Computer Vision Models against Common Corruptions (Arxiv 23)
>
> [7] PoseTrack21: A Dataset for Person Search, Multi-Object Tracking and Multi-Person Pose Tracking (CVPR 21)
>
> [8] Divide and Conquer: Hybrid Pre-training for Person Search (AAAI 24)
>
> [9] PS-ARM: An End-to-End Attention-aware Relation Mixer Network for Person Search (ACCV 22)
>
> [10] Ground-to-Aerial Person Search: Benchmark Dataset and Approach (MM 23)
>
> [11] SAT: Scale-Augmented Transformer for Person Search (WACV 23)

---

> ### Author Response · Authors · 2024-11-26
> **A gentle reminder for reviewer-author discussion**
>
> Dear reviewer Yohg,
>
> As the reviewer-author discussion period is coming to a close, we kindly ask if there are any remaining concerns or points about our submission that we haven't sufficiently addressed. We're ready to provide additional clarifications or information if needed.
>
> Once again, we appreciate your valuable efforts and feedback to strengthen our work.
>
> Best regards,
>
> Authors of Submission 7041

---

> ### Comment · Reviewer_Yohg · 2024-11-27
>
> Thanks for your reply. What concerns me is that the "real-world" dataset in your work contains only 400 samples, which is far from sufficient to represent the full diversity of real-world scenarios. As a result, many potential corruptions that exist in practical applications are not covered in the benchmarks, raising questions about the generalizability of the proposed method. Apparantly, more real-world evaluation should be conducted. Also, the computational overhead of the method has not been discussed, which could pose significant challenges when deploying in real-time applications.

---

> > ### Author Response · Authors · 2024-12-02
> > **A gentle reminder for reviewer-author discussion**
> >
> > Dear Reviewer Yohg,
> >
> > As the reviewer-author discussion period is coming to a close, we kindly ask if there are any other remaining concerns or points about our work that we haven't sufficiently addressed in the rebuttal. We're ready to provide additional clarifications or information.
> >
> > We are grateful for your time and efforts to strengthening our work.
> >
> > Best regards,
> >
> > Authors of Submission 7041

---

> ### Author Response · Authors · 2024-11-27
>
> Dear Reviewer Yohg,
>
> Thank you for your constructive feedback and for taking time to visit our rebuttal discussion.
>
> ---
>
> ### **About the computational overhead of our method when deployed in real-time applications**
>
> Our method does **NOT** introduce any computational cost in real-time applications, **as both our foreground-aware augmentation and regularization are training techniques**.
>
> Without additional model parameters or real-time cost, our method can be seamlessly integrated into existing person search models, as shown in Tab 3 of the main paper.
>
> ---
>
> ### **About the scalability of validation result requiring more real-world evaluation**
>
> We understand your concern about the scalability of the result since the real domain dataset we construct are not extensively large.
>
> Due to the requirement of obtaining multiple viewpoints of the same person, inherent ethical issues, and privacy concerns, extensive data collection in person search demands substantial costs with careful consideration, with these costs escalating further when extended to special domains such as corrupted environments.
>
> In this context, for some challenging conditions of this field, small size datasets[12,13], which only include (150 / 600) and (420 / 600) number of images (#test split / #entire dataset), have contributed to advance the field.
>
> Given these aforementioned challenges in this field, our approach of constructing corruption benchmarks can serve as an effective proxy for measuring robustness across diverse environments.
>
> Since collecting new real-world data confronts rapidly escalating costs when attempting to cover comprehensive corruption scenarios, our framework offers the effective solution in this aspect, where the approach of establishing corruption environments within our framework has been extensively demonstrated across numerous computer vision fields [14,15,16,17,18,19,20].
>
> Furthermore, our frequency sensitivity analysis in Figure 6 of the main paper demonstrates that applying our method to the baseline shows improving resilience to a wide range of frequency perturbations including low-, middle- and high-frequency.
> Yin et al. (2019) analyzed the common corruptions in the frequency domain, demonstrating that fog and contrast have relatively low-frequency components, noise-related corruptions have high-frequency components, and blur and pixelate have middle-frequency components. Given this comprehensive coverage of our method across the frequency spectrum, we believe our approach to be adaptable to unknown corruptions in other frequency ranges as well.
>
> —
>
> References:
>
> [12] Cross-Domain Adversarial Feature Learning for Sketch Re-identification (MM 18)
>
> [13] Partial Person Re-identification (ICCV 15)
>
> [14] When Human Pose Estimation Meets Robustness: Adversarial Algorithms and Benchmarks (CVPR 21)
>
> [15] Robustness Disparities in Face Detection (NeurIPS 22)
>
> [16] Benchmarking Robustness of 3D Object Detection to Common Corruptions in Autonomous Driving (CVPR 23)
>
> [17] RoboDepth: Robust Out-of-Distribution Depth Estimation under Corruptions (NeurIPS 23)
>
> [18] Benchmarking Robustness of Adaptation Methods on Pre-trained Vision-Language Models (NeurIPS 23)
>
> [19] Robo3D: Towards Robust and Reliable 3D Perception against Corruptions (ICCV 23)
>
> [20] Benchmarking the Robustness of Cross-view Geo-localization Models (ECCV 24)

---

### Official Review · Reviewer_tUry · 2024-11-02

**Soundness:** 4
**Presentation:** 4
**Contribution:** 3
**Rating:** 8
**Confidence:** 4

**Summary:**

Previous studies have only investigated the corruption issues in person detection or person re-identification separately, whereas this paper is the first to study the corruption robustness in the person search task (detection + re-identification). The authors meticulously constructed two benchmarks in corruption scenarios, CUHK-SYSU-C and PRW-C, and through extensively evaluating models on these two benchmarks, they obtained some meaningful findings. Subsequently, based on these findings, the authors proposed several methods to improve the model's robustness against corruption, and validated the effectiveness of these methods through extensive experiments. Overall, this is a very solid work and I like it.

**Strengths:**

1. The writing of this paper is smooth and easy to understand. The authors structured it by first posing the problem, then constructing the benchmarks, evaluating current models to identify areas for improvement, proposing improvement methods, and finally validating the effectiveness of the methods. I am able to follow the core ideas of this work very well.

2. According to the authors, they are the first to study the issue of corruption robustness in the person search task, a problem which I believe has significant practical relevance. In real-world applications, due to the influence of weather, lighting, and other imaging conditions, the sursurveillance videos are prone to corruption. A person search model that can withstand such corruptions is essential.

3. The contributions of this paper are substantial. The paper introduces two corrupted benchmarks CUHK-SYSU-C and PRW-C (as well as an additional manual annotated test set based on BDD100K), and proposes two modules to enhance the model's resistance to corruption, all of which have positive implications for the development of this field.

4. The paper provides rich details in its experimental procedures, offering good reproducibility.

**Weaknesses:**

Although the entire work is very solid, it relatively lacks novelty at the technical level. Through evaluation and analysis, the authors concluded that foreground corruption and robust person representation are important. Therefore, they proposed two methods: Foreground-Aware Augmentation and Regularization for Robust Person Representation. I carefully reviewed the technical details of these two methods and found them to be somewhat lacking in innovation, as they are based on existing techniques that have already been implemented.

**Questions:**

1. Will you make your entire code, model, and dataset public? I believe this would be significant for the development of the field.

2. Since you pointed out that person re-identification representation is important, have you tried to investigate the effect of some person reid pre-trained models on the robustness of the person search model against corruptions?  To my knowledge, pre-training can learn good person re-identification representations, and I would be very interested to see an experiment where you investigate the impact of pre-trained person representations on your task. For instance, you could conduct an experiment with the pre-trained ResNet50 in LUP, UPReID, LUP-NL, ISR, PLIP and CION.

I will greatly appreciate it and **raise the score** if you could conduct the aforementioned experiment that interests me.

Reference:

LUP: Unsupervised pre-training for person re-identification (CVPR2021)

UPReID:  Unleashing potential of unsupervised pre-training with intra-identity regularization for person re-identification (CVPR2022)

LUP-NL: Large-scale pre-training for person re-identification with noisy labels (CVPR2022)

ISR: Identity-seeking self-supervised representation learning for generalizable person re-identification (ICCV2023)

PLIP: Plip: Language-image pre-training for person representation learning (NeurIPS2024)

CION: Cross-video Identity Correlating for Person Re-identification Pre-training (NeurIPS2024)

---

> ### Author Response · Authors · 2024-11-20
>
> Dear Reviewer tUry,
>
> Thank you for your insightful comments and positive remarks on our paper's structure and contributions. We appreciate your feedback and the requested interesting experiments. We have elaborated on the points or questions raised in your review below.
>
> ---
> ### **Question about code, model, and dataset availability**
>
> Yes! Like mentioned in the paper, we will release our complete framework including code, model, and dataset.
>
> ---
>
> ### **Question about additional experiments**
>
>
> | Initialization Method | CUHK-SYSU-C |  | CUHK |  |
> |---|---|---|---|---|
> | | rR@1 / rmAP | R@1 / mAP | R@1 / mAP | |
> | ImageNet pretrained | 46.2 / 44.4 | 43.6 / 41.6 | 94.5 / 93.8 | |
> | LUP pretrained | 49.3 / 47.6 | 46.4 / 44.5 | 94.1 / 93.5 | |
> | LUPNL pretrained | 54.4 / 52.2 | 51.8 / 49.4 | 95.3 / 94.7 | |
> | PLIP pretrained | 50.8 / 49.0 | 46.7 / 44.7 | 91.8 / 91.4 | |
> | CION pretrained | 53.2 / 51.5 | 50.4 / 48.5 | 94.6 / 94.1 | |
>
> | Initialization Method | PRW-C |  | PRW |  |
> |---|---|---|---|---|
> | | rR@1 / rmAP | R@1 / mAP | R@1 / mAP | |
> | ImageNet pretrained | 46.7 / 22.6 | 38.9 / 10.5 | 83.4 / 46.7 | |
> | LUP pretrained | 46.8 / 23.0 | 39.0 / 10.7 | 83.3 / 46.5 | |
> | LUPNL pretrained | 51.7 / 27.4 | 44.5 / 14.1 | 86.1 / 51.5 | |
> | PLIP pretrained | 49.1 / 25.2 | 40.2 / 11.2 | 81.8 / 44.3 | |
> | CION pretrained | 54.8 / 29.7 | 47.1 / 15.6 | 85.9 / 52.5 | |
>
> We appreciate your insight to suggest the interesting experiment, which also interests us.
>
> We conduct the experiment with pretrained methods mentioned above.
>
> As you anticipated, in most cases, pretrained initializations with good person reID representations contribute to measurable improvements, which also aligns with our findings.
> Thanks to your suggested experiment, it confirms the potential of enhanced representation ability from pretrained knowledge to improve robustness, a more salient approach to enhance the representation utilizing external knowledge (e.g. [1]) can be valuable to explore than current naïve utilization (model initialization).
>
> In our experiment, we utilize pretrained ResNet50 models from LUP, LUP-NL, PLIP, CION, while we were unable to include UPReID and ISR in our current rebuttal timeline due to code availability issues or missing ResNet weights.  Like our paper's experimental setup, we use SeqNet as the baseline model and conduct experiments on CUHK-SYSU and PRW datasets.
>
> We will revise our paper to reflect additional experiments and discussions.
>
> ---
>
> ### **Although the entire work is very solid, it relatively lacks …**
>
> Thank you for your careful review and insightful feedback on our work.
>
> We would like to emphasize that one of our main contributions is the first pioneering analysis on this topic, and providing a comprehensive framework where the evaluations, analyses, and further methodological research can be encouraged.
>
> Our method, though relatively simple in its design, derives from our robustness analysis and observations of the person search framework, showing to be simple-yet-effective in its performance. Also, its simplicity enables its broad applicability across various person search models, as shown in Table 3 (main paper). With the advantage that our method maintains clean set performance within a ±2% range, it pushes the robustness performance of diverse state-of-the-art models to a large margin, where the models themselves show a limited resilience to corruptions in Table 1 (main paper).
>
> Even though we leverage some existing techniques such as AugMix, we have designed our method tailored to the person search task, based on our experimental observations. To the best of our knowledge, we are the first to introduce selective augmentation and its corresponding regularization approach in the person search field.
>
> Nonetheless, we also agree that future technique-focused research, such as advanced network architectures for robust person search, can also be conducted, which we believe is a promising direction for future work. We believe our comprehensive framework, benchmarks, and analyses will serve as a valuable foundation for such future technical innovations in the field.
>
> ---
>
> References:
>
> [1] Making person search enjoy the merits of person re-identification (PR 2022)

---

> > ### Comment · Reviewer_tUry · 2024-11-21
> >
> > Thank you for your detailed reply and conducting the experiments that I'm interested in. I hope these experiments can be added to the revised version of your paper. Even though it lacks a bit of innovation on the technical level, the work in this paper is solid on the whole if it is open-sourced. I think this paper has reached the acceptance level of ICLR. I will raise my score to 8.

---

> ### Author Response · Authors · 2024-11-21
>
> Dear Reviewer tUry,
>
> Thank you for taking the time to consider our responses and for raising the score.
> We are pleased that our responses have answered your questions.
>
> Your interesting experimental suggestion has enabled us to improve the paper and verify the findings of our work.
> We will include the additional experimental results and discussions in our revision.
>
> If you have any remaining questions or feedback, we would be glad to provide additional information or results.
> We appreciate your insightful review and your time in improving our work.
>
> Best regards,
>
> Authors of submission 7041

---

### Official Review · Reviewer_d5hd · 2024-11-02

**Soundness:** 3
**Presentation:** 3
**Contribution:** 2
**Rating:** 3
**Confidence:** 3

**Summary:**

This paper explores the robustness of person search models in degraded data environments. It introduces two new benchmark datasets (CUHK-SYSU-C and PRW-C) for evaluating the performance of person search models under different degraded conditions. By analyzing the sensitivity of both the detection and feature extraction stages under corruption, the authors found that existing models exhibit significant vulnerability when foreground images are damaged. Based on these findings, a foreground-aware data augmentation and regularization method was proposed to enhance model robustness.

**Strengths:**

This paper is the first to investigate the robustness of person search models in degraded data environments. It introduces two novel benchmark datasets (CUHK-SYSU-C and PRW-C) for evaluating the performance of person search models under various degradation scenarios.

**Weaknesses:**

1. The strength of this paper lies in the introduction of two new benchmark datasets for exploring person search in interference scenarios. However, overall, the dataset construction methods are largely adapted from existing approaches, and the proposed solution is relatively simple, focusing on narrowing the feature gap between corrupted and original images. From a theoretical innovation standpoint, the contributions are limited and fall short of ICLR's high standards.

2. While the foreground-aware augmentation approach is effective, it appears to be an incremental improvement rather than a fundamentally new concept. The paper could be strengthened by deeper theoretical insights into why this method improves robustness beyond empirical results.

3. The paper employs various existing data augmentation methods to simulate real-world disturbances. However, the key question is whether these augmentations can effectively benefit real-world scenarios. If their impact is limited, the constructed datasets may hold little significance.

**Questions:**

1. The experiments demonstrate that foreground damage has a greater impact on model performance. However, it is unclear whether this damage affects the detection performance more or the recognition performance.

---

> ### Author Response · Authors · 2024-11-24
> **Official Comment by Authors (1/2)**
>
> Dear Reviewer d5hd,
>
> Thank you for your thorough review and constructive feedback. We appreciate your time and effort reviewing our work. Below, we address your concerns.
>
> ---
>
> ### **About real-world relevance of our corruptions**
>
> |Real-world Validation|Fog|Dark|Rain|Blur|
> |---|---|---|---|---|
> |**Training data \ Metric**|R@1/mAP|R@1/mAP|R@1/mAP|R@1/mAP|
> |CUHK-SYSU|56.8/47.8|33.3/34.7|31.6/35.9|64.0/64.0|
> |CUHK-SYSU-C|**65.0/55.9**|**40.3/43.2**|**37.7/42.7**|**67.8/67.1**|
>
> About data augmentation(corruption) to simulate real-world disturbances,
> during the rebuttal period, we conduct additional experiments to further validate the effectiveness of our corruption benchmark.
> Instead of using clean data for the model training, we use our corruption benchmark for model training, and evaluate the performance on the four corruption scenarios -fog, dark, rain and blur- in BDD100K, a real-world corruption dataset as introduced in Tab 5 of the main paper. The experimental results show that training with our corruption benchmark helps improve the performance, verifying that our benchmark can serve as an effective proxy for the real world by bridging the distribution gap.
>
> cf. About data augmentations used in our method,
> our method shows performance improvement on real-world corruption scenarios, as validated in Table 5 of the main paper.
>
> ---
>
> ### **About foreground-aware augmentation**
>
> During the rebuttal period, we provide a deeper exploration of our foreground-aware augmentation with the qualitative analysis and Grad-CAM visualization, as shown in the Figure 8b of Appendix B.5.
>
> We compare the content in bounding box regions after augmentation applied based on two regional criteria: entire image (1st column) and foreground region (2nd column).
>
> Naive adoption of augmentation by applying it to the entire image (1st column) could lead to unreliable bounding boxes, as observed in the 1st column where bounding boxes miss some discriminative parts of a person.
>
> In contrast, with foreground-aware strategy (2nd column), most discriminative parts of the person remain within the bounding box while augmentation is successfully applied. Preserving discriminative information within the bounding box is crucial, as it serves as the criterion for selecting region proposals used in training person search models. Accordingly, the results of foreground-aware strategy in 5th column show that the model better captures the person's discriminative information (4th/5th column: without/with applying foreground-aware strategy).
>
> Data augmentation has proven to be an effective approach in various computer vision fields,
>
> But the adoption of data augmentation in person search remains relatively unexplored, as the naïve use could lead to the problem as shown in the 1st column of Figure 8b (Appendix) or Figure 5 (Main paper), such as severe semantic perturbation and unreliable bounding boxes, which lead us to pioneer the application of data augmentation for robustness in person search.
>
> Although this technique might seem a bit incremental, we pioneer the usage of data augmentation by tailoring it to the person search task, based on our experimental observations.
>
> To the best of our knowledge, we are the first to introduce the selective augmentation strategy in the person search field.
>
> This technique, though relatively simple in its design, by leveraging our observations, shows simple yet effective performance. It has the advantage of broad applicability for various person search models without introducing additional parameters or inference time cost.
>
> ---
>
> ### **Question about our experimental observations**
> Thank you for your insightful question.
>
> We had measured both detection and recognition performance, as illustrated in Figure 4a of the main paper. As shown in the figure, the recognition performance experiences more significant degradation compared to detection performance.
>
> Furthermore, we would like to refer to Appendix A.3, which provides a clear answer to your question.
>
> We simultaneously conduct two experiments that led to our observations: Individual evaluation of detection and reID stage, and evaluation where corruption is applied to either the background or the foreground, which can be categorized into four testing scenarios:
> - fg-corrupted & Representation against Corruption
> - fg-corrupted & Detection against Corruption
> - bg-corrupted & Representation against Corruption
> - bg-corrupted & Detection against Corruption
>
> The results show that ***this experiment also exhibits consistent results with our observations***, leading to the conclusion that ***the representation stage is particularly vulnerable when damage is applied to the foreground as well***.

---

> ### Author Response · Authors · 2024-11-24
> **Official Comment by Authors (2/2)**
>
> ### **About design principles of our benchmark and method (Answer for ‘The strength of this paper lies in …’)**
>
> #### **About the benchmark:**
> In the field of corruption robustness, the prevalent approach is to evaluate the model performance by adding noise to the image, which has contributed to establish the framework for robustness analysis across various computer vision fields (Yi et al., 2021; Chen et al., 2021; Kong et al., 2023), and it encourages continuous exploration in this direction [1,2, Lee et al., 2022].
>
> Although our framework leverages some existing techniques, ***its design is specialized with task-specific insights for evaluating person search framework***, beyond merely employing the existing approach.
>
> - **Our benchmark construction is guided by experimental observation.** Based on the findings in Appendix A.1, we apply different corruptions to query and gallery images. As demonstrated in Appendix Table 6, applying identical corruptions to both query and gallery images could introduce bias by artificially increasing image similarity. This design choice also reflects the insight that query and gallery images may not always exhibit the same type or severity level of corruption in real-world scenarios.
>
> - **We carefully select corruption types and calibrate their severity levels optimized for evaluating person search framework.** In deciding types of corruption, we focus on real-world deployment scenarios while maintaining comprehensive coverage of corruption types. Given that noise-related corruptions in ImageNet-C have an overlapping score close to 1 [3], we replace shot-noise and impulse-noise with rain and dark corruptions, which better represent realistic deployment scenarios. We calibrate severity levels to be challenging while ensuring that persons in the images remain detectable and re-identifiable by humans even at the highest severity. In Appendix C.3, we provide detailed documentation of all control factors and their corresponding parameters used to define severity levels for all 18 corruptions.
>
> - **Mitigating ethical concerns of new data collection in person search**, our benchmark serves as a proxy for diverse scenarios . As the person search field demands careful ethical consideration due to its use of human data, our methodology provides an analysis framework containing diverse corruption scenarios without ethical concerns of new data collection.
>
> We would like to emphasize that even though we leverage the existing technique, our benchmarks derive the first pioneering analysis on this topic, thus providing a comprehensive framework where the evaluations, analyses, and further methodological research can be encouraged.
>
> #### **About the proposed solution:**
> Our proposed solution, though relatively simple in its design, derives from our robustness analysis and observations of the person search framework, showing to be simple-yet-effective in its performance. Also, its simplicity enables its broad applicability across various person search models without incurring additional parameter or inference time costs. With the advantage that our method maintains clean set performance within a ±2% range, it pushes the robustness performance of diverse state-of-the-art models to a large margin.
>
> —
>
> References:
>
> [1] Robust Heterogeneous Federated Learning under Data Corruption (ICCV 23)
>
> [2] Towards Better Robustness against Common Corruptions for Unsupervised Domain Adaptation (ICCV 23)
>
> [3] Using the Overlapping Score to Improve Corruption Benchmarks (ICIP 21)

---

> ### Author Response · Authors · 2024-11-25
> **A gentle reminder for reviewer-author discussion**
>
> Dear Reviewer d5hd,
>
> As the reviewer-author discussion period is coming to a close, we kindly ask if there are any remaining concerns or points about our submission that we haven't sufficiently addressed. We're ready to provide additional clarifications or information if needed.
>
> Once again, we appreciate your valuable efforts and feedback to strengthen our work.
>
> Best regards,
>
> Authors of submission 7041

---

> > ### Comment · Area_Chair_3nhm · 2024-11-27
> > **Rebuttal Update**
> >
> > Dear Reviewer d5hd,
> >
> > Could you please check the reviewer comments from other reviews and the feedback from the authors? This paper receives divergent scores, your comment would be important to make a fair decision.
> >
> > Thanks,
> > Your AC

---

> ### Comment · Reviewer_d5hd · 2024-11-27
>
> I have reviewed the comments from the other reviewers, and overall, the proposed new task in this paper is of practical significance. However, there are still several issues that need to be addressed, which is why I have decided to lower my score:
>
> 1. **Limited Contribution of the Dataset**: The proposed dataset is simply an augmented version of an existing dataset and does not accurately simulate real-world scenarios. In fact, the test dataset should be sourced from real-world environments in order to effectively validate the actual performance of the proposed method.
>
> 2. **Limited Innovation of the Method**: Both I and reviewers Yohg and tUry pointed out in our initial evaluations that the innovation of the proposed method is relatively weak. Furthermore, since the dataset does not simulate real-world conditions, the performance validation of the proposed method is insufficient.
>
> 3. **Inconsistent Reviewer Evaluations**: Reviewer ujfi gave the paper a perfect score and claimed there are no obvious weaknesses, which clearly contradicts the opinions of the other reviewers. For example, reviewer Yohg pointed out that the manuscript contains numerous typographical errors, which are basic issues.
>
> Overall, while I believe the proposed new task has practical significance, the authors' efforts have not sufficiently supported this task, and as a result, I am inclined to lower my score. The authors should further improve the construction of the dataset and the design of the method.

---

> > ### Author Response · Authors · 2024-12-02
> > **A gentle reminder for reviewer-author discussion**
> >
> > Dear Reviewer d5hd,
> >
> > As the reviewer-author discussion period is coming to a close, we kindly ask if there are any other remaining concerns or points about our work that we haven't sufficiently addressed in the rebuttal. We're ready to provide additional clarifications or information.
> >
> > We are grateful for your time and efforts to strengthening our work.
> >
> > Best regards,
> >
> > Authors of Submission 7041

---

> ### Author Response · Authors · 2024-11-29
> **Official Comment by Authors (1/2)**
>
> Dear Reviewer d5hd,
>
> We appreciate your constructive feedback and for taking time to visit our rebuttal discussion.
>
> ---
>
> ### **Multifaceted reviewer guidance has largely strengthen our work (Answer for 3)**
>
> We have addressed the typographical ambiguities and incorporated these changes in the revised paper. We remain open to enhancing notations and clarifications if necessary, and would appreciate any future suggestions.
>
> On the other hand, **we are pleased that our presentation has facilitated easy understanding for readers**, as recognized by positive comments from other reviewers such as "solid work" (tUry) and "easy to understand" (tUry, Yohg, ujfi).
>
> **Thanks to all reviewers for taking the time to improve our work, our work has become** ***even stronger***,
> - by providing robustness analyses of additional six recently proposed models (Reviewer FKyP),
> - by verifying the effectiveness of proposed method on three recent models (Reviewer Yohg),
> - by validating the scalability of our framework (Reviewer Yohg),
> - by validating the effectiveness of the proposed corruptions (Reviewer FKyP),
> - by verifying the validity of our observations with the experiments in Tab 7 and with pretraining (Reviewers d5hd, tUry),
> - by further exploring our method's intuition through CAM analysis (Reviewer d5hd).
>
> Aligning with Reviewer ujfi's insight, we believe that our pioneering endeavor and framework, grounded in our motivations and analyses acknowledged by Reviewer FKyP, will serve as a starting point and stimulate future explorations in corruption robustness, including real world data collection and advanced methodology.
>
> ---
>
> ### **About the value of our method (Answer for 2)**
>
> *Before starting, we would like to notify that the development of our robustness solution was made possible thanks to the foundations for robustness evaluation and analyses our benchmarks provide.*
>
> We understand your concern that its design of our method might appear relatively simple.
>
> Though our solution might look simple, it was derived from our observations of the person search framework, leading to show a simple yet effective performance.
>
> Thanks to its simplicity, it can be seamlessly applied to person search models without worrying about additional parameters or real-time costs. Consequently, our method can significantly strengthen the robustness of various existing well-designed person search models, where the models themselves show a limited resilience to corruptions in Table 1 of the main paper.
>
> Even though we leverage some existing techniques such as AugMix, we have designed our method tailored to the person search task, based on our experimental observations. As a result, our method can strengthen the robustness while maintaining clean set performance within ±2% range.
>
> We provide ***evidence in various ways validating the generalizability*** of our method.
> - **We validate the generalizability of our method across four real world corruption scenarios**, as shown in Table 6 of the main paper.
> - **Our frequency sensitivity analysis** in Figure 6 of the main paper demonstrates that applying our method to the baseline shows *improving resilience to a wide range of frequency perturbations including low-, middle- and high-frequency*. Yin et al. (2019) analyzed the common corruptions in the frequency domain, demonstrating that fog and contrast have relatively low-frequency components, noise-related corruptions have high-frequency components, and blur and pixelate have middle-frequency components. Given this comprehensive coverage of our method across the frequency spectrum, we believe our approach to be adaptable to unknown corruptions in other frequency ranges as well.
> - **Extensive experiments with five different person search models** validate the effectiveness of our proposed method.
>
> Nonetheless, we also agree that future technique-focused research, such as advanced network architectures for robust person search, can also be conducted, which we believe is a promising future direction. We believe our comprehensive framework, benchmarks, and analyses will serve as a valuable foundation for such future technical innovations in the field.

---

> ### Author Response · Authors · 2024-11-29
> **Official Comment by Authors (2/2)**
>
> ### **About our benchmarks not sourced from actual real environment (Answer for 1)**
>
> We understand your concern that we should source the test dataset from the real world to support the proposed new task. Indeed, real-world data collection has its own advantages.
>
> In the current landscape, when we want to validate if our person search algorithm is robust in the real world, we need to pay high costs in order to collect the test dataset, while confronting unique challenges inherent to the data person search requires: privacy issues. This challenge becomes more complex when attempting to cover various real-world scenarios. Furthermore, gathering such diverse real-world scenarios from various locations and data collection environments becomes infeasible.
>
> In this context, ***our framework provides a cost-effective, diverse covering, scalable solution, which is free from privacy issues***.
>
> Our approach of establishing corruption environments by curating the existing data ***has been extensively demonstrated across numerous computer vision fields [4,5,6,7,8,9,10]***.
>
> Moving beyond simply augmenting, our proposed benchmarks are specially tailored with task-specific insights for evaluating person search framework, as outlined in the above comment.
>
> As shown in the experiment of the above comment, our benchmark can narrow the distribution gap towards the real world, ***validating it to serve as an effective proxy for the real world***.
>
> - **Diverse scenario coverage**: Due to the high costs of real data collection, comprehensive scenario coverage remains limited in this field. To our best knowledge, no existing real dataset encompasses as many as 18 different scenarios. In contrast, our framework provides 18 arrays of corruption scenarios, firstly enabling robustness measurement of person search models across diverse corruption environments, encouraging robustness evaluations, analyses, and further methodological improvements.
> - **Scalable**: When collecting the corruption scenarios through real world data collection, each collection becomes dependent on the specific situation of the location where it was collected. However, our framework is scalable and can be combined with other existing datasets collected in various ways from different situations, as verified in the experiment with larger scale PoseTrack21[11] dataset, conducted by the request of Reviewer Yohg.
> - **Avoiding privacy issues**: As the person search field demands careful ethical consideration due to its use of human data, our methodology provides an analysis framework containing diverse corruption scenarios without introducing privacy concerns of new data collection. It implies that *introducing our benchmark constructing approach to the person search field is especially valuable*.
>
> Although our approach may not perfectly mirror real-world environments, we believe our approach brings a meaningful research value by providing cost-effective, comprehensive, scalable, ethics-issue-free solutions to the field. As our work is the beginning of the branch of corruption robustness in person search, we believe our approach will serve as good preliminary work for future research, including future real-world data collection endeavors.
>
> ---
>
> References:
>
> [4] When Human Pose Estimation Meets Robustness: Adversarial Algorithms and Benchmarks (CVPR 21)
>
> [5] Robustness Disparities in Face Detection (NeurIPS 22)
>
> [6] Benchmarking Robustness of 3D Object Detection to Common Corruptions in Autonomous Driving (CVPR 23)
>
> [7] RoboDepth: Robust Out-of-Distribution Depth Estimation under Corruptions (NeurIPS 23)
>
> [8] Benchmarking Robustness of Adaptation Methods on Pre-trained Vision-Language Models (NeurIPS 23)
>
> [9] Robo3D: Towards Robust and Reliable 3D Perception against Corruptions (ICCV 23)
>
> [10] Benchmarking the Robustness of Cross-view Geo-localization Models (ECCV 24)
>
> [11] PoseTrack21: A Dataset for Person Search, Multi-Object Tracking and Multi-Person Pose Tracking (CVPR 21)

---

### Official Review · Reviewer_FKyP · 2024-11-05

**Soundness:** 2
**Presentation:** 3
**Contribution:** 2
**Rating:** 8
**Confidence:** 4

**Summary:**

This paper focus on the corruptions of images in person search problems. Two benchmarks are proposed for evaluation, while the analysis is adopted on both detect and reID stages. A method with data augmentation and regularizaiton is further proposed. Good experimental resutls are achieved.

**Strengths:**

1. The motivaion of evaluationg the affects of image corruption for person search is useful.
2. The analysis results and discussion about the performance dropping in both detection and searching stages are acceptable.

**Weaknesses:**

1. It is hard to regard as a contribution via simply post-processing two exisiting datsasets. The human-designed or generated augmentation is far from the real scene.
2. The further proposed forground-aware augmentation is not novel, which has been a common sense that the key region for accurate person search is the body region.
3. The referred methods in the experiments are most out-of-date, missing some recently proposed methods.

**Questions:**

See weakness.

---

> ### Author Response · Authors · 2024-11-23
> **Official Comment by Authors (1/2)**
>
> Dear Reviewer FKyP,
>
> Thank you for your insightful reviews and for taking the time to evaluate our work. Below, we address your concerns.
>
> ---
> ### **About extended evaluation with recently proposed methods**
>
> During the rebuttal period, we evaluate the corruption robustness of ***six recently proposed models***, hoping that our work will contribute to the development of this field.
>
> We will include the additional experimental results in our revision.
>
> |Method|CUHK|CUHK-C|CUHK-C|PRW|PRW-C|PRW-C|
> |---|---|---|---|---|---|---|
> ||R@1/mAP|rR@1/rmAP|R@1/mAP|R@1/mAP|rR@1/rmAP|R@1/mAP|
> |AlignPS [1]|93.1/92.6|48.3/46.3|45.0/42.8|82.0/45.4|45.9/22.3|37.7/10.1|
> |AlignPS+ [1]|94.4/93.8|**51.6**/**49.3**|**48.7/46.3**|82.4/45.6|47.8/24.7|39.3/11.3|
> |PS-ARM [2]|94.8/94.1|50.2/48.5|47.5/45.6|85.2/52.0|**52.4**/**27.2**|**44.7**/**14.1**|
> |HKD [3]|94.9/94.2|48.8/47.0|46.4/44.3|85.1/51.5|49.8/24.4|42.4/12.5|
> |SAT [4]|94.8/94.4|47.3/45.3|44.8/42.7|**87.5**/**54.5**|51.1/24.6|**44.7**/13.4|
> |RoI-AlignPS [5]|**96.0**/**95.4**|43.5/41.4|41.8/39.5|84.0/50.3|51.1/25.3|42.9/12.7|
>
> ---
>
> ### **About foreground-aware augmentation**
>
> Thank you for your careful review and insightful feedback on our work.
>
> Although this might seem as common sense, we are the first to leverage this insight as methodological application in the person search field, to our best knowledge.
>
> The adoption of data augmentation in person search remains relatively unexplored, as the naïve use can lead to the problem as shown in Fig 4 main paper, such as severe semantic perturbation and unreliable bounding boxes, which lead us to pioneer the application of data augmentation for robustness in person search.
>
> This technique, though relatively simple in its design, is derived from our robustness analysis and observations of the person search framework, which makes it simple yet effective.
>
> Also, its simplicity enables its broad applicability across various person search models without incurring additional parameter or inference time costs.

---

> ### Author Response · Authors · 2024-11-23
> **Official Comment by Authors (2/2)**
>
> ### **About the value of our benchmark beyond dataset processing**
>
> |Testing Domain (Real-world)|Fog|Dark|Rain|Blur|
> |-------------------------|------------|------------|------------|------------|
> |**Training Domain \ Metric**|R@1/mAP|R@1/mAP|R@1/mAP|R@1/mAP|
> |Clean|56.8/47.8|33.3/34.7|31.6/35.9|64.0/64.0|
> |Our corruption|**67.5**/**58.0**|**42.5**/**43.9**|**39.4**/**43.0**|**68.5**/**67.6**|
>
> In the field of corruption robustness, the prevalent approach is to evaluate the model performance by adding noise to the image, which has contributed to establish the framework for robustness analysis across various computer vision fields (Yi et al., 2021; Chen et al., 2021; Kong et al., 2023), and it encourages continuous exploration in this direction [6,7, Lee et al., 2022].
>
> Although our framework leverages some existing techniques, ***its design is specialized with task-specific insights for evaluating person search framework***, beyond merely employing the existing approach.
>
> - **Our benchmark construction is guided by experimental observation.** Based on the findings in Appendix A.1, we apply different corruptions to query and gallery images. As demonstrated in Appendix Table 6, applying identical corruptions to both query and gallery images could introduce bias by artificially increasing image similarity. This design choice also reflects the insight that query and gallery images may not always exhibit the same type or severity level of corruption in real-world scenarios.
>
> - **We carefully select corruption types and calibrate their severity levels optimized for evaluating person search framework.** In deciding types of corruption, we focus on real-world deployment scenarios while maintaining comprehensive coverage of corruption types. Given that noise-related corruptions in ImageNet-C have an overlapping score close to 1 [8], we replace shot-noise and impulse-noise with rain and dark corruptions, which better represent realistic deployment scenarios. We calibrate severity levels to be challenging while ensuring that persons in the images remain detectable and re-identifiable by humans even at the highest severity.
>
> - **While avoiding ethical concerns in person search**, our benchmark serves as a proxy for diverse scenarios . As the person search field demands careful ethical consideration due to its use of human data, our methodology provides an analysis framework containing diverse corruption scenarios without ethical concerns of new data collection.
>
> We would like to emphasize that even though we leverage the existing technique, our benchmarks derive the first pioneering analysis on this topic, thus providing a comprehensive framework where the evaluations, analyses, and further methodological research can be encouraged.
>
> During the rebuttal period, we conduct additional experiments to further validate the effectiveness of our corruptions. We conduct the experiments on four corruption scenarios: dark, rain, blur, and fog. We first train our model on each proposed corruption corresponding to four scenarios, subsequently, we evaluate the model on BDD100K, a real-world corruption dataset as referred to in Tab 5 of the main paper.
>
> The experimental results show that our proposed corruptions can bridge the distribution gap with real corruption scenarios, ***serving as an effective proxy for the real world***.
>
> ---
>
> ### References
>
> [1] Efficient Person Search: An Anchor-Free Approach (IJCV 23)
>
> [2] Ground-to-Aerial Person Search: Benchmark Dataset and Approach (MM 23)
>
> [3] PS-ARM: An End-to-End Attention-aware Relation Mixer Network for Person Search (ACCV 22)
>
> [4] SAT: Scale-Augmented Transformer for Person Search (WACV 23)
>
> [5] Anchor-Free Person Search (CVPR 21)
>
> [6] Robust Heterogeneous Federated Learning under Data Corruption (ICCV 23)
>
> [7] Towards Better Robustness against Common Corruptions for Unsupervised Domain Adaptation (ICCV 23)
>
> [8] Using the Overlapping Score to Improve Corruption Benchmarks (ICIP 21)

---

> ### Comment · Reviewer_FKyP · 2024-11-25
>
> Thanks for the detailed response! My previous concerns are well addressed and I would like to raise my rating to accept.

---

> > ### Author Response · Authors · 2024-11-25
> >
> > Dear Reviewer FKyP,
> >
> > Thank you for your thoughtful consideration and for raising the score. We are glad that our responses have well addressed your concerns.
> >
> > If you have any remaining questions or feedback, we would be glad to provide additional clarifications or results if needed. Please don't hesitate to let us know.
> >
> > We sincerely appreciate your valuable feedback to improve our work.
> >
> > Best regards,
> >
> > Authors of submission 7041

---

### Author Response · Authors · 2024-12-04
**Final Global Response by Authors**

Dear reviewers and chairs,

We appreciate all reviewers for their reviews and efforts during the rebuttal.

We are glad that the reviewers found our work "unexplored aspects or good motivation" (FKyP, d5hd, tUry, ujfi), "novel benchmarks or assisting future development" (d5hd, tUry, ujfi), "useful analysis for the field" (ujfi, FKyP), "easy to understand" (Yohg, tUry, ujfi), "extensive experiments" (Yohg, tUry, ujfi).
- **Our benchmarks provide the foundation to stimulate further research** in this area, by pioneering an unexplored aspect of person search.
- **We measure the robustness of 12 seminal person search models**, providing valuable insights for the person search community.
- **The analyses and observations** can guide improvements in related fields, with their validity confirmed by consistent results from two experiments conducted during the rebuttal.
- **Our method addresses the robustness vulnerabilities revealed in seminal models**, providing substantial improvements to their performance.
- **We are pleased by positive feedback on our presentation**, such as  'easy to understand', 'logically clear', 'able to follow the core ideas well', which has been further improved by reviewers' valuable efforts during the rebuttal.

---

We are pleased that our responses have addressed concerns and received positive feedback with increased ratings, which has been very encouraging us.

To summarize the points we have addressed:

>### **The value of our task-specialized benchmarks: Beyond traditional real-world new data collection**
1. Although our framework leverages some existing techniques, ***its design is specialized with task-specific insights for evaluating person search framework***, beyond merely employing the existing approach. (Please refer to our response for Reviewer FKyP for more detail)
   - During the rebuttal, we have shown the value of our benchmark that it can bridge the distribution gap towards the real world.
2. Compared to collecting new data from actual real world, our approach offers distinct advantages: ***Cost-effective, Diverse covering, Scalable, and Privacy-issue-free***, where our approach of establishing corruption environments has been extensively demonstrated to serve as a solid foundation across numerous computer vision fields.
   - **Diverse covering**: While the cost becomes easily infeasible when attempting to cover various real-world scenarios
, our framework 18 diverse scenarios.
   - **Scalable**: While the data collection from real world becomes dependent on the specific situation where it was collected, our framework is scalable, leveraging various collecting environments of other datasets, as verified with larger scale dataset conducted by the request of Reviewer Yohg.
   - **Privacy-issue-Free**: As the person search demands careful ethical consideration, introducing our approach of constructing benchmark is especially beneficial for this field.

>### **Simple-yet-effective baseline for robust person search**

Preface: *The birth of our robust person search baseline was possible thanks to the foundations for robustness framework our benchmarks provide. We hope our method, alongside our benchmark, will encourage future endeavors.*

Our method, though relatively simple in its design, derives from our robustness analysis and observations of the person search framework, showing to be simple-yet-effective result.

Also, its simplicity enables its broad applicability across various person search models. While keeping clean set performance within ±2%, it pushes the robustness to a large margin.

Even though we leverage some existing techniques, we have designed our method tailored to the person search task, based on our experimental observations.

We provide the verification of our robustness solution regarding the scalability, generalizability and deeper analysis: extensive validation across diverse models (Tab 3); generalizable towards real corruption (Tab 5); CAM analysis (Fig 8).

>### **About scalability of validation result of our method requiring more real-world evaluation**

We understand Reviewer Yohg's main remaining concern that real-world validation result of our method might not be scalable due to potential unknown corruptions. To address this, we had analyzed our method from a frequency sensitivity perspective.

The Fig 6 demonstrates that applying our method to the baseline shows improving resilience to a wide range of frequency perturbations including low-, middle- and high-frequency. Yin et al. (2019) analyzed the common corruptions in the frequency domain, revealing that fog and contrast have relatively low-frequency components, noise-related corruptions have high-frequency components, blur and pixelate have middle-frequency components. Given this comprehensive coverage of our method across the frequency spectrum, we believe our approach to be adaptable to unknown corruptions in other frequency ranges as well.

Sincerely,

Authors of submission 7041

---

### Author Response · Authors · 2024-12-04
**Remarks for Final Consideration: Comment by Corresponding Author**

Dear Reviewers and Area Chairs,

Thank you for the thoughtful feedback and valuable discussions throughout the review process. We deeply appreciate the time and
effort you have dedicated to evaluating our work.

We acknowledge the limitations of our study, as noted by some reviewers. Specifically, our benchmarks rely on augmentations and lack
the scale and diversity of large, real-world datasets. Developing such datasets would require substantial resources and significant time
investment—likely one to two years or more.

Nevertheless, we believe our contribution is an important step forward in addressing robustness challenges in person search. As highlighted by reviewers, our work introduces a novel perspective by focusing on the robustness of end-to-end person search under corruptions, a critical yet underexplored area. The benchmarks we propose, while not exhaustive, provide a foundation for evaluating and understanding the vulnerabilities of existing and follow-on models.

It is worth noting that many existing techniques underperformed on our benchmark, not due to intrinsic shortcomings but because they were not designed with such scenarios in mind. The absence of similar benchmarks or evaluation protocols in the past has limited the
development of robustness-aware methods. By addressing this gap, we aim to spark innovation in this direction.

We hope that by sharing our current dataset with the research community, we can catalyze efforts to develop more robust techniques
for person search. These efforts will, in turn, pave the way for the creation of larger and more diverse real-world datasets, driving the
field forward in the coming years.

Once again, we sincerely thank the reviewers for their constructive comments, which have greatly improved the quality of our work. We are committed to incorporating this feedback and contributing further to this important area of research.

Best regards,

Corresponding Author

---

### Meta-Review · Area_Chair_3nhm · 2024-12-20

**Metareview:**

This paper received two negative and three positive reviews. Positive reviews praised the paper's valuable exploration of the impact of image corruption on person search tasks, especially its relevance to real-world surveillance affected by weather and lighting. The authors introduce two corrupted benchmarks (CUHK-SYSU-C and PRW-C) and propose modules to enhance model robustness, addressing gaps in previous datasets and evaluation methods. The paper is well-structured, clear, and offers original insights into foreground and background effects, making a notable contribution to person search research. However, the negative reviews raised several concerns: the proposed benchmark lacks detailed severity levels, and the dataset mainly augments existing ones, limiting its originality. The module offers only incremental improvements, with minimal novelty, and has not been validated on larger datasets, questioning its scalability. Additionally, key related works are missing, and there are typographical errors. Overall, while the benchmark introduces some new elements, it lacks significant theoretical innovation and does not fully simulate real-world scenarios, limiting its applicability.
After reviewing the authors’ response, the Area Chair (AC) remains concerned that the benchmarks are not sourced from real-world environments and that the scalability of the method has not been adequately addressed. Therefore, the Area Chair recommends rejecting the paper.

**Additional Comments On Reviewer Discussion:**

This paper receive divergent reviews. Five reviewers have joined the discussion. After reviewing the authors’ response and the manuscript, the Area Chair (AC) remains concerned that the benchmarks are not sourced from real-world environments and that the scalability of the method has not been adequately addressed. I agree this paper has some merits, but the quality of this paper should be further improved to meet the bar for ICLR.

---

### Decision · Program_Chairs · 2025-01-22

Reject